



# Nitrate transboundary heavy pollution over East Asia in winter

Syuichi Itahashi[1], Itsushi Uno[2], Kazuo Osada[3], Yusuke Kamiguchi[3#], Shigekazu Yamamoto[4], Kei Tamura[5], Zhu Wang[2], Yasunori Kurosaki[6], Yugo Kanaya[7]

[1] Environmental Science Research Laboratory, Central Research Institute of Electric Power Industry, 1646 Abiko, Abiko, Chiba, 270-1194, Japan
[2] Research Institute for Applied Mechanics, Kyushu University, 6-1 Kasuga Park, Kasuga, Fukuoka, 816-8580, Japan
[3] Graduate School of Environmental Studies, Nagoya University, D2-1 (510) Furo-cho, Chikusa-ku, Nagoya, Aichi, 464-8601, Japan
[4] Fukuoka Institute of Health and Environmental Sciences, 39 Mukaizano, Dazaifu, Fukuoka, 818-0135, Japan
[5] Nagasaki Prefectural Environmental Affairs Department, 2-1306-11 Ikeda, Omura, Nagasaki, 856-0026, Japan
[6] Arid Land Research Center, Tottori University, 1390 Hamasaka, Tottori, 680-0001, Japan
[7] Japan Agency for Marine-Earth Science and Technology, 3173-25 Showa-machi, Kanazawa-ku, Yokohama, Kanagawa, 236-0001, Japan
[#]Present address: JCB Co. Ltd., Tokai Regional Office, 2-16-26, Nishiki, Naka-ku, Nagoya, Aichi 460-0003, Japan

*Correspondence to*: Syuichi Itahashi (isyuichi@criepi.denken.or.jp)

**Abstract.** High $PM_{2.5}$ concentrations reaching around 100 μg/m$^3$ were observed twice during an intensive observation campaign in January 2015 at Fukuoka (33.52°N, 130.47°E) in western Japan. These events were analyzed comprehensively by a regional chemical transport model and synergetic ground-based observations with state-of-the-art measurement systems, which can capture the behavior of secondary inorganic aerosols ($SO_4^{2-}$, $NO_3^-$, and $NH_4^+$). The first episode was dominated by $NO_3^-$ (type N), whereas the second episode was dominated by $SO_4^{2-}$ (type S). The concentration of $NH_4^+$, which is the counterion for $SO_4^{2-}$ and $NO_3^-$, was high for both types. The sensitivity simulation of the chemical transport model showed that the dominant contribution was from transboundary air pollution for both types. To investigate the differences between these types of transboundary heavy pollution further, the chemical transport model results were examined in combination with the backward trajectory analysis. The air mass originated from northeast China and reached Fukuoka for both types, but the traveling time from the coastline of China to Fukuoka was 18 h for type N and 24 h for type S. The conversion ratio of $SO_2$ to $SO_4^{2-}$ (Fs) was less than 0.1 for type N, but reached 0.3 for type S as the air mass approached Fukuoka. The higher Fs for type S was related to the higher relative humidity and concentration of $HO_2$, which produces the most effective oxidant, $H_2O_2$, for the aqueous-phase production of $SO_4^{2-}$. Analyzing the gas ratio, which is an indicator of the sensitivity of $NO_3^-$ to changes in $SO_4^{2-}$ and $NH_4^+$, showed that the air mass over China was super $NH_3$-rich for type N, but was almost $NH_3$-neutral for type S. Higher $NO_3^-$ concentrations were maintained during transport owing to the lower $SO_4^{2-}$ for type N, whereas the production of $SO_4^{2-}$ led to decomposition of $NH_4NO_3$ and more $SO_4^{2-}$ was transported for type S. The transboundary air pollution dominated by $SO_4^{2-}$ in type S is a major acid transport process over East Asia. However, our study confirms the importance of the transboundary air pollution dominated by $NO_3^-$ (type N), which will help refine our understanding of the transboundary heavy $PM_{2.5}$ pollution in winter over East Asia.





## 1 Introduction

Particulate matter (PM) presents major environmental problems globally, especially in East Asia. A typical example is the episode of severe air pollution that occurred in January 2013 above China (e.g., Wang et al., 2014; Uno et al., 2014). During this episode, PM with aerodynamic diameters of less than 2.5 μm ($PM_{2.5}$) reached record-breaking concentrations of 772

μg/m$^3$ on January 12 (Pan et al., 2016). The transboundary air pollution in downwind regions resulting from the severe air pollution in China is also an important environmental problem. For example, the possible long-range transport of $PM_{2.5}$ was based on the comparison of observations in metropolitan areas and remote islands in western Japan (Kaneyasu et al., 2014). They highlighted the dominant effect of the transboundary transport for sulfate ($SO_4^{2-}$) as a major $PM_{2.5}$ component in western Japan throughout most of the year. During spring, due to the prevailing westerly wind over East Asia, transboundary

air pollution of both aerosols and gases (e.g., carbon monoxide (CO) and ozone ($O_3$)) have been discussed thoroughly (Itahashi et al., 2010; 2013; 2015; Kanaya et al., 2016; Nagashima et al., 2010). In summer, the clean air mass from oceans is moved over Japan by the southerly wind caused by the Pacific High; however, some studies have discussed the importance of transboundary air pollution from China over western Japan (Itahashi et al., 2012a; Ikeda et al., 2014). Recently, one-year source-receptor relationships for $SO_4^{2-}$ were evaluated, and China was identified as the main influence on downwind regions

throughout the year, with local sulfur dioxide ($SO_2$) emissions making an important contribution during summer (Itahashi et al., 2016). Compared with the analyses for spring and summer, transboundary air pollution events during winter are less well understood.

In this study, we also focused on nitrate ($NO_3^-$), which is an important $PM_{2.5}$ component. $NO_3^-$ is produced via the reaction of gas-phase nitrate (nitric acid; $HNO_3$) and ammonia ($NH_3$), and this process is reversible. This reaction favors a shift toward

the aerosol phase at low temperatures and high humidity (Seinfeld and Pandis, 2006). The simulated spatial distribution over East Asia showed the possible impact of the transboundary pollution of $NO_3^-$ during winter over western Japan (Zhang et al., 2007; Ying, 2014). However, a quantitative evaluation over downwind regions was not presented in their studies. This is partly because model ability was not evaluated owing to the difficulty in measuring $NO_3^-$. Particulate $NH_4NO_3$ may be volatilized after collection on the filter, by either an increase in the pressure drop across the particle-collecting medium or

changes in the gas-aerosol equilibrium during sampling (Sickles et al. 1999; Chang et al., 2000). This volatilization could occur even in winter because the temperature in the instrument shelter can be increased by heat from the pump. Therefore, the ground-based Acid Deposition Monitoring Network in East Asia uses the four-stage filter pack method: $NH_4NO_3$ is collected on the first filter and gas-phase $HNO_3$ and $NH_3$ are detected on the subsequent filters. The artifacts might not be significant; however, to avoid the possibility of volatilization, total nitrate (the sum of $NO_3^-$ and $HNO_3$) has been used to

evaluate the model ability in previous studies (e.g., Kajino et al., 2011).

To improve our understanding of the behavior of $NO_3^-$, accurate measurements and the evaluation of model ability are needed. In this study, we used the state-of-the-art automated monitoring system for $SO_4^{2-}$ and $NO_3^-$, Aerosol Chemical Speciation Analyzer (ACSA). This system measures $SO_4^{2-}$ and $NO_3^-$ with high temporal resolution and 1 h intervals were




used in this study, minimizing the possibility of volatilization. In addition, the behavior of ammonium ($NH_4^+$), which is the counterion for $SO_4^{2-}$ and $NO_3^-$, was captured by the well-validated NHx monitoring system. Therefore, the secondary inorganic aerosols (sulfate ($SO_4^{2-}$)−nitrate ($NO_3^-$)−ammonium ($NH_4^+$); SNA) were fully observed by our synergetic monitoring system. The denuder-filter pack (D-F pack) method with 6 h cycles was also used during the intensive

observation period from January 7–17, 2015 to support and validate the ACSA and NHx monitoring system. Based on these measurement systems, gas-phase $HNO_3$ and $NH_3$ can be measured by the D-F pack method and NHx monitor, respectively. The related gas-phase behavior analysis is valuable for improving our understanding of the formation of $NO_3^-$. The observations were conducted at the Chikushi Campus of Kyushu University, which is in the suburbs of Fukuoka City (33.52°N, 130.47°E) in western Japan. The synergetic ground-based observation dataset was systematically interpreted by

using the regional chemical transport model, and we also examined the impact of the domestic and transboundary air pollution during winter. The chemical transport model studies are one of a critical approach to analyze three-dimensional air pollutants behavior and to estimate source impacts. The systematical comparison of model results with observations including gas-phase precursors will promote our understanding for model ability in Asia. This will also contribute to the Model Inter-Comparison Study for Asia (MICS-Asia) which focuses on a common understanding of model performances

and uncertainties and especially on long-range transport in Asia (Carmichael et al., 2002; 2008, Li et al., 2015). This paper is constructed as follows. Section 2 documents the observation dataset and model simulation. Section 3 discusses the results with respect to temporal variations at observation sites and the model results combined with the backward trajectory analysis. Finally, a summary and conclusions are given in Section 4.

**2 Observation and model simulation**

**2.1 Observation sites**

The synergetic observations for capturing SNA behavior were conducted at Chikushi Campus of Kyushu University located in the suburbs of Fukuoka City (33.52°N, 130.47°E). Fukuoka City is the largest center of commerce in Kyushu Island. The population of Fukuoka City is 1.5 million and that of the Fukuoka metropolitan area is 2.5 million. This is the fourth largest metropolitan area in Japan, after Tokyo (34.8 million), Osaka (12.2 million), and Nagoya (5.5 million). This site is an urban

site. In addition to the observation at Fukuoka, observations from the Goto Islands (32.68°N, 128.83°E) and Tsushima Island (32.20°N, 129.28°E) were also used. The Goto Islands are located in the East China Sea, 190 km southwest of Fukuoka, and they have a population of 70,000. Tsushima Island is located in the Tsushima straits, 140 km northwest of Fukuoka, and has a population of 34,000. These two islands have negligible anthropogenic emissions sources and are regarded as remote sites. In addition to these three sites over Kyushu island, to investigate the regions affected by transboundary air pollution,

observations from Tottori City (35.54°N, 134.21°E) in western Japan were also used. Tottori City has a population of 190,000, and this site is also regarded as a remote site in western Japan. The locations of these four observation sites over Japan are shown in Fig. 1. In addition to these observations over Japan, $PM_{2.5}$ observations in China from the US Embassy in





Beijing and the US Consulates in the provincial capitals of Shanghai and Shenyang were used. The locations of these three sites over China are shown in Fig. 2.

### 2.1.1 Aerosol chemical speciation analyzer

An ACSA-12 Monitor (Kimoto Electric Co., Ltd., Osaka, Japan), for $PM_{10}$ and $PM_{2.5}$, which were separated by a US Environmental Protection Agency inlet and a virtual impactor, were measured with high temporal resolution (Kimoto et al., 2013). PM was collected on a tape filter made of Teflon (PTFE). Hourly observations were conducted for $SO_4^{2-}$ and $NO_3^-$ at Fukuoka. The mass concentrations of PM were determined by using the beta-ray absorption method. The ACSA-12 measured $NO_3^-$ using an ultraviolet spectrophotometric method, and $SO_4^{2-}$ by turbidimetry after addition of $BaCl_2$ to form $BaSO_4$ and polyvinyl pyrrolidone as a stabilizer. The analytical period was within 2 h of sampling; therefore, the volatilization of particulate $NH_4NO_3$ after collection was regarded as small compared with the traditional filter-pack observation method. ACSA has been tested (Osada et al., 2016) and used to analyze the severe winter haze in Beijing (Zheng et al., 2015; Li et al., 2016), and to identify the aerosol chemical compositions at Fukuoka (Pan et al., 2016)

### 2.1.2 NHx monitor

The behaviors of $NH_3$ and $NH_4^+$ are also important because they are the counterions for $SO_4^{2-}$ and $NO_3^-$. The concentrations of gaseous $NH_3$ and aerosol $NH_4^+$ were measured with a semi-continuous microflow analytical system (Kimoto Electric Co. Ltd., MF-NH$_3$A, Osada et al., 2011) at Fukuoka. The atmospheric NHx was dissolved in ultrapure water with a continuous air–water droplet sampler and quantified by fluorescence (excitation, 360 nm; emission, 420 nm) of the *o*-phtalaldehyde–sulfite–$NH_3$ reaction product (Genfa et al., 1989). Two inlet lines were used to differentiate the total amounts of NHx and particulate $NH_4^+$ after gaseous $NH_3$ was removed by a phosphoric acid-coated denuder from the sample air stream. The cut-off diameter of the inlet impactor was about 2 μm.

### 2.1.3 Denuder-filter pack method

During the intensive observation period from January 7–17, 2015, D-F pack measurements were conducted at Fukuoka to validate the ACSA and NHx monitoring measurement systems. An annular denuder–multi-stage filter sampling system was used for $HNO_3$ and size-segregated aerosol sampling. The sampling interval was 6–8 h. At the inlet, coarse-mode aerosols were removed by Nuclepore membrane filters (111114, Nomura Micro Science Co., Ltd., Atsugi, Japan, pore size: 8 μm), and then gas-phase $HNO_3$ was collected with the annular denuder (2000-30x242-3CSS, URG Co.) coated with NaCl (Perrino et al., 1990). Fine-mode aerosols were collected with a PTFE filter (J100A047A, ADVANTEC, Tokyo, Japan, pore size: 1 μm), and a nylon filter (66509, Pall Co.) captured volatized nitrate from the PTFE filter (Appel et al., 1981; Vecchi et al., 2009). The sample air flow rate was 16.7 L/min (1 atm, 25 °C). Under these conditions, the aerodynamic diameter of 50% cut-off for the Nuclepore filter was about 1.9 μm (John et al., 1983). The samples were analyzed by ion chromatography (IC). Comparing the size-segregated $SO_4^{2-}$ and $NO_3^-$ data based on the D-F pack method with ACSA showed systematic





differences. Fine-mode aerosols were underestimated by the D-F pack compared with the ACSA $PM_{2.5}$ measurements, and coarse-mode aerosols were overestimated by the D-F pack compared with the ACSA $PM_c$ measurements, because of difference in the cut-off diameter for the D-F pack method (Osada et al., 2016). Details of the comparison and validation of the ACSA data are reported in Osada et al. (2016).

### 2.1.4 PM-712

Hourly $PM_{10}$ and $PM_{2.5}$ concentrations were measured by a PM monitor (PM-712, Kimoto Electric Co., Ltd.) at the Goto Islands, Tsushima Island, and Tottori. The ionic constituents of the species on the PTFE tape filters were also analyzed by IC to compare the aerosol behaviour at Fukuoka and the other sites. At all sites, sample spots collected on the tape filter were covered with polyester tape to avoid contamination and crosstalk interference during storage. The sampling duration for the PM-712 tape filters was 1 h, except for Tottori where it was 3 h. For chemical analysis, four consecutive 1 h tape samples were combined into one sample for the Goto Islands and Tsushima Island. For Tottori, tape samples for 1 or 0.5 days were combined. Because of a temperature change during PM sample storage, some $NO_3^-$ may have escaped via volatilization of $HNO_3$ from the sample. Therefore, $NO_3^-$ data from the Goto Islands, Tsushima Island, and Tottori were not used. Hereafter, we call these datasets from the PM tape samples 'tape filter'.

### 2.1.5 Beta attenuation monitors

Hourly $PM_{2.5}$ concentrations in China were measured by beta attenuation monitors (BAMs) (1020, MetOne Instruments, Inc., Grants Pass, OR, USA) at the U.S. Embassy in Beijing from April 8, 2008, at the U.S. Consulates in Shanghai from December 21, 2011, and at the U.S. Consulates in Shenyang from April 22, 2013 (MOE, 2016). In the BAM technique, PM is collected on a quartz filter tape over a given time interval and the attenuation of beta rays through the sample is measured and correlated directly with the PM mass. The details and the statistical analysis results of the BAM observation at the U.S. Embassy and Consulates are found in the work of San Martini et al. (2015).

### 2.1.5 Multi-angle absorption photometer

Observations of black carbon (BC), a primary aerosol that directly reflects local emissions contributions, from Fukuoka and the Goto Islands were also used to distinguish domestic and transboundary air pollution. BC is observed by using a multi-angle absorption photometer (MAAP; MAAP5012, Thermo Fisher Scientific, Waltham, MA, USA) (Petzold et al., 2005). In this method, the absorbance of the particles deposited on the filter is distinguished from scattering by reflectance measurements at multiple angles and by transmittance. This is to minimize the effects of coexisting aerosol particles other than BC on filter-based absorption photometers. The comparison measurements of BC from the Goto Islands were previously performed by Kanaya et al. (2013), and they reported that the BC MAAP measurements were strongly correlated with measurements by other techniques but had a positive bias. From the results reported by Kanaya et al. (2013), the MAAP





absorption cross section of 6.6 $m^2/g$ was systematically increased to 10.3 $m^2/g$ at 639 nm. There were no MAAP BC measurements from the Goto Islands before January 11, 2015 during the intensive observation period.

## 2.2 Chemical transport model

The chemical transport model simulation was performed by using the Community Multi-scale Air Quality (CMAQ)
modeling system version 4.7.1 (Byun and Schere, 2006) with nesting over East Asia. The meteorological fields of CMAQ were prepared with the Weather Research and Forecasting model version 3.3.1 (Skamarock et al., 2008) with analysis nudging applied to the National Centers for Environmental Prediction final operational global analysis data. The model domain covers the whole of East Asia with an 81 km horizontal grid resolution, with a 95 × 75 grid centered at 30°N and 115°E on a Lambert conformal projection. The nested domain covers eastern China and the whole of Japan with a 27 km
horizontal grid resolution and a 145 × 145 grid. The vertical grid for sigma-pressure coordinates extends to 50 hPa with 37 layers with nonuniform spacing. Lateral boundary condition was prepared by global chemical transport model of Geos-Chem (Uno et al., 2014). The simulation period was from January 1–17, 2015, and the first 6 days were discarded as model spin-up time. The dry deposition velocity of $HNO_3$ over land was increased by a factor of five based on the model intercomparison results (Shimadera et al., 2014; Morino et al., 2015).
Emissions were set as follows. Anthropogenic emissions and natural sources of NOx from soil were obtained from the latest Regional Emission inventory in ASia (REAS) version 2.1 (Kurokawa et al., 2013), which covers 2000 to 2008. Therefore, the emissions for January 2008 were used in this study. This assumption is based on the following reasons. Satellite observation of the $NO_2$ column showed a decreasing trend in NOx emissions from China of -6%/year after 2011, and the levels for 2015 are similar to those for 2009 (Irie et al., 2016). In contrast, the $NO_2$ column over Japan decreased until 2013,
and then began to increase from 2013 owing to the change in power plant use after the Fukushima Daiichi nuclear disaster (Morino et al., 2011). The level of the $NO_2$ column over Japan in 2015 was close to that in 2008 (Irie et al., 2016). The installation of flue-gas desulfurization systems in power plants in China decreased $SO_2$ emissions in China from 2005 to 2006; after this turning point, the variation in $SO_2$ emissions is complicated (Xia et al., 2016). In Japan, $SO_2$ emissions are increasing for the same reasons as for NOx; however, there are no reliable references for the current status of $SO_2$ emissions.
Considering these factors in the variation of NOx and $SO_2$ emissions over China and Japan, it was assumed that the 2008 emissions would be within the range of uncertainty of the bottom-up emission inventories. Because REAS does not consider the monthly variation in $NH_3$, we used the monthly variation estimated by Huang et al. (2013). Biogenic emissions were prepared from MEGAN (Model of Emissions of Gases and Aerosols from Nature) (Guenther et al., 2012). Biomass burning emissions were used from the climatological database of RETRO (REanalysis of the TROpospheric chemical composition)
(Schulrz et al., 2008). Volcanic activity data were taken from ACESS (Ace-Asia and TRACE-P Modeling and Emission Support System) (Streets et al., 2003) and were modified by the volcanic activity observation data from the Japan Meteorological Agency (JMA) for available volcanoes (JMA, 2016). Fourteen main active volcanoes in Japan and Mt. Mayon and Mt. Bulsan on Luzon Island in the Philippines were considered. The model simulation using the above dataset is





referred to as the base case simulation. The modeling system domain with overlaid anthropogenic NOx emissions is shown in Fig. 3.

To investigate whether the effect of domestic or transboundary air pollution is dominant, we also conducted a sensitivity simulation, in which the anthropogenic emissions in Japan are switched off. Shipping emissions were not treated in the sensitivity simulation. Because the amount of emissions from China is larger than that from Japan, to avoid large nonlinearities in the atmospheric concentration response to emissions variation (e.g., Itahashi et al., 2015), the sensitivity simulation was designed to switch off the anthropogenic emissions in Japan. Based on the differences between the base case simulation and this sensitivity simulation, the domestic contribution from Japan was estimated.

### 3 Results and Discussion

#### 3.1 Meteorological conditions

Meteorological conditions during the intensive observation campaign from January 7–17, 2015 are shown in Fig. 2 with observations and model results. Meteorological observation stations of the JMA in the corresponding nested model grid of Fukuoka were used. Temperatures (Fig. 2a) were around 5 °C at night and 10 °C during the day in January 2015. On January 9, the temperature was nearly 0 °C at Fukuoka. For the wind field, because of the dominance of the northwesterly wind system from the Asian continent during winter, the wind direction was generally 270°–360° (west to north) and the wind speed was around 5 m/s, with the exception of January 9 and 12–15. On January 9, when the coldest temperature during the intensive observation campaign was observed, the wind speed was less than 1 m/s and wind direction was from the south. On January 13–15, the wind speed was also low, at 2–3 m/s, and the wind direction was easterly, caused by a warm front passing over the south of Kyushu Island on January 14. After the warm front had passed, the relative humidity was close to 100% on January 15–16, with a maximum of 10 mm/h rain on January 15. Comparing the observations with the model results shows that our modeling system generally captures the observed meteorological variations during this episode.

#### 3.2 Temporal variation of particulate matter

(a) PM$_{2.5}$

The temporal variation of PM$_{2.5}$ over Japan at Fukuoka, Tsushima Island, the Goto Islands, and Tottori are presented in Fig. 1. PM$_{2.5}$ observation data are taken from ACSA at Fukuoka, and are taken from PM-712 at other sites. Temporal resolutions are 1 h for all observations. During the analyzed period of January 7–17, 2015, episodic PM$_{2.5}$ peaks reached around 100 μg/m$^3$ at Fukuoka twice. The first peak, observed at 12:00 LT on January 11 (shown in blue in Fig. 1) reached a maximum concentration of 86.4 μg/m$^3$ at Fukuoka and 105.1 μg/m$^3$ at the Goto Islands. During this first peak, the concentration at Tsushima Island was 63.9 μg/m$^3$, which was lower than at the other remote island site in the Goto Islands, and there was no distinctive peak at Tottori. The second peak observed at 00:00 LT on January 17 (shown in red in Fig. 1) reached a maximum concentration at Fukuoka of 106.2 μg/m$^3$. During this second peak, the remote sites of the Goto Islands and





Tsushima Island also recorded high $PM_{2.5}$ concentrations of 104.8 and 89.1 μg/m$^3$, respectively, and the $PM_{2.5}$ concentration reached 37.6 μg/m$^3$ at Tottori. In Fig. 1, we show the model results as black lines. Generally, the model captured the observed temporal $PM_{2.5}$ behavior, although it underestimated the first peaks at Fukuoka and the Goto Islands, and the second peak at the Goto Islands. The timing of the high $PM_{2.5}$ concentration was reproduced well by the modeling system.

Statistical analysis of the model reproducibility demonstrated that all paired datasets for $PM_{2.5}$ showed good correlations between the observations and the model at the four sites in Japan, with a correlation coefficient (R) of 0.86. Mean fractional bias (MFB) and mean fractional error (MFE) were -42.6% and 67.4%, respectively, and these results satisfied the model performance criteria (MFB ≤ ±60% and MFE ≤ +75%) proposed by Boylan and Russell (2006). Figure 1 also shows the model results of a sensitivity simulation performed by switching off the Japanese anthropogenic emissions. The sensitivity

simulations results are shown as dotted black lines and the difference between the base case and the sensitivity simulation is shown in gray, which indicates the domestic contribution of Japan. Except for Fukuoka, there were little domestic contributions for $PM_{2.5}$; therefore, the transboundary air pollution was dominant during January 2015. At Fukuoka, although domestic contributions for $PM_{2.5}$ were found in some cases on January 8–10 and 14, the concentration of $PM_{2.5}$ was lower compared with the two peaks. During the two episodes when $PM_{2.5}$ concentration reached around 100 μg/m$^3$ over Japan, the

model simulation suggested that the effect of transboundary air pollution was dominant, even at Fukuoka.

The temporal variations of $PM_{2.5}$ over China at Beijing, Shanghai, and Shenyang are shown in Fig. 2. $PM_{2.5}$ observation data are taken from BAM-1020 and the temporal resolution is 1 h. At Beijing, there were high concentrations of $PM_{2.5}$ which correspond to the high concentration of $PM_{2.5}$ found over Japan. One high concentration was approximately 300 μg/m$^3$ on January 10–11, and other was around 600 μg/m$^3$ on January 16. These peak times were almost one day before the high

concentration was observed over Japan. At Shanghai, there were two clear peaks with a $PM_{2.5}$ concentration of 200 μg/m$^3$ on January 11 and 17. The time corresponded well with the peak time over Japan. At Shenyang, where the local emissions from domestic sources was dominant in winter, the temporal variation was complex compared with Beijing and Shanghai. $PM_{2.5}$ showed sharp peaks several times with concentrations of around 300 μg/m$^3$, whereas the model only showed gentle peaks. Analysis of the model reproducibility showed that all $PM_{2.5}$ paired datasets for the observations and model at three sites over

China showed that R was 0.73, and MFB and MFE were -9.8% and 46.8%, respectively; satisfying the model goal criteria (MFB ≤ ±30% and MFE ≤ ±50%) proposed by Boylan and Russell (2006). The evaluation of model performance over China supports the discussion of downwind regions.

(b) SNA

The temporal variations of SNA are shown in Figs. 5 and 6. In Fig. 5, $SO_4^{2-}$ and $NH_4^+$ are shown for four sites in Japan. At Fukuoka, ACSA and D-F pack observations are shown for $SO_4^{2-}$, and NHx monitor and D-F pack results are shown for $NH_4^+$. The temporal resolutions of ACSA and the NHx monitor were 1 h, and those of the D-F packs were 6–8 h depending on the samples. For the Goto Islands, Tsushima Island, and Tottori, the PM-712 tape filter data were used. The temporal resolution was 4 h at the Goto Islands and Tsushima Island, and 1 or 0.5 day at Tottori. In Fig. 6, $NO_3^-$, $HNO_3$, $NH_3$, and total ammonia





(sum of $NH_4^+$ and $NH_3$) are shown for Fukuoka. ACSA and D-F pack observations for $NO_3^-$ are shown, D-F pack observations are shown for $HNO_3$, and NHx monitor observations are shown for $NH_3$ and total ammonia. Because of a temperature change during PM-712 sample storage, $NO_3^-$ concentrations could have been affected by volatilization; hence, only $NO_3^-$ analysis at the Fukuoka site was used. At Fukuoka, the SNA concentration contributed 52% and 46% of the $PM_{2.5}$

concentration in the first and second episodes, respectively. For $SO_4^{2-}$ (Fig. 5 (left)), the concentration during the second episode was larger than during the first episode at Fukuoka, the Goto Islands, and Tsushima Island. At Tottori, there was no peak for the first episode for $SO_4^{2-}$. In contrast to $SO_4^{2-}$, a higher $NO_3^-$ concentration was observed during the first episode instead of the second episode (Fig. 6). $NH_4^+$ showed high concentrations during both episodes because it is the counterion for $SO_4^{2-}$ and $NO_3^-$ (Fig. 5 (right)). Based on the analysis of the $PM_{2.5}$ (Fig. 3) and SNA (Figs. 5 and 6) observations, the $PM_{2.5}$

concentrations were similar during the episodes on January 11 and 17; however, the main component of SNA was $NO_3^-$ during the first episode and $SO_4^{2-}$ during the second episode. Therefore, the first episode (shown in blue in Figs. 3, 5, and 6) is referred to as 'type N', and the second episode (shown in red in Figs. 3, 5, and 6) is referred to as 'type S' hereafter.

The model results for the base case and sensitivity simulations are overlaid with the same temporal resolution as the observations in Figs. 5 and 6. The model tended to underestimate the $SO_4^{2-}$ concentration (Fig. 5 (left)); however, the model

reproduced the features of types N and S, and sensitivity simulation indicated the dominance of transboundary air pollution for $SO_4^{2-}$ during the intensive observation campaign in January 2015, even at Fukuoka. For $NO_3^-$ (Fig. 6), the model reproduced the features of the type N and S peaks well, although the model overestimated the dip in $NO_3^-$ concentration found from the evening of January 10 to before the type N episode. The D-F pack observations generally underestimated $NO_3^-$ compared with the ACSA observations, because of the difference in cut-off diameter between these measurement

systems. Except for the type N and S episodes, domestic contributions were seen for $NO_3^-$ on January 8–10 and 14. However, the sensitivity simulation confirmed that the transboundary $NO_3^-$ air pollution was dominant for types N and S. Because $NH_4^+$ is the counterion for both $SO_4^{2-}$ and $NO_3^-$, small domestic contributions for $NH_4^+$ were observed at Fukuoka (Fig. 5 (right)). This result corresponded to the domestic contribution for $NO_3^-$. For the other three remote sites, there were no domestic contributions for $NH_4^+$.

The behavior of SNA, gas-phase $HNO_3$, and $NH_3$ were analyzed comprehensively based on the NHx monitor and D-F pack observations (Fig. 6) to support our understanding of $NO_3^-$ behavior. There are few synergetic analyses including gas-phase behavior over the downwind region of Asian continent. Peaks for gas-phase $HNO_3$ were found for types N and S, whereas the concentration of gas-phase $NH_3$ was nearly zero (less than $1\mu g/m^3$ for 24 h average) for types N and S, and on January 8, 10, 12, and 15 (green arrows in Fig. 6). The concentration of total ammonia showed distinct peaks for types N and S;

therefore, the nearly zero concentration of $NH_3$ suggested the full conversion of $NH_3$ to produce $NH_4^+$ as a counterion for $SO_4^{2-}$ and $NO_3^-$. The sensitivity simulation, in which Japanese anthropogenic emissions were switched off, clarified the different features of related gas-phase species. The base case simulation and sensitivity simulation were similar for $HNO_3$, suggesting it originated from transboundary air pollution. A slight increase in $HNO_3$ in the sensitivity simulation was found on January 8–10 and 12 (red arrows in Fig. 6). These were the complex cases connected to overseas and domestic emissions.



If there are no Japanese $NH_3$ emissions, the transported $HNO_3$ cannot produce $NO_3^-$ in Japan, and so it remains as gas-phase $HNO_3$. The synergetic analysis for gas-phase $HNO_3$ and $NH_3$ indicated that abundant $HNO_3$ was transported from abroad and reacted with domestic $NH_3$, producing $NO_3^-$ on January 8–10. Compared with these cases, domestic $HNO_3$ and $NH_3$ produced $NO_3^-$ on January 14 (orange arrows in Fig. 6). The concentrations were lower than for type N and S, which were

dominated by the transboundary air pollution.

 (c) Coarse mode aerosols

Coarse-mode aerosols were also partly analyzed in this study. Because of the effect of transboundary air pollution on $HNO_3$ (Fig. 6), we focused on coarse-mode $NO_3^-$. Coarse-mode $NO_3^-$ is produced by reactions of $HNO_3$ with mineral dust or sea-

salt particles. In general, mineral dust mainly has an effect in spring over East Asia, whereas sea-salt particles play an important role throughout the year. Recently, we reported the importance of coarse-mode $NO_3^-$ as an atmospheric input in East Asian ocean regions (Itahashi et al., 2016). Figure 7 shows the modeled and observed coarse-mode $NO_3^-$, $Na^+$, and $Cl^-$. ACSA and D-F pack observations are shown for coarse-mode $NO_3^-$, and D-F pack observations are shown for coarse-mode $Na^+$ and $Cl^-$. During the intensive observation period in January, coarse-mode $NO_3^-$ also showed high concentrations for

types N and S of around 10 $\mu g/m^3$, and on January 9–10 of around 5 $\mu g/m^3$. Based on the model results, because the domestic contribution for $HNO_3$ was observed on January 14 (Fig. 6), the domestic contribution for coarse-mode $NO_3^-$ was observed only on January 14, but the concentration was below 1 $\mu g/m^3$. $Na^+$ and $Cl^-$ from sea-salt particles also had peaks for types N and S. Sea-salt particles are mechanically produced by high winds; therefore, these peaks generally corresponded to high wind speeds (Fig. 2). High winds were observed on January 15, and $Na^+$ and $Cl^-$ peaks occurred, but the coarse-mode

$NO_3^-$ concentration was close to zero. This was because there was no $HNO_3$ to react with NaCl from January 12–15, and wet deposition of coarse-mode $NO_3^-$ with precipitation from noon on January 14 to the evening of January 15. For coarse-mode $NO_3^-$, transboundary air pollution was the dominant factor. This means that a large amount of $HNO_3$ was transported from abroad (Fig. 5) and reacted with sea-salt particles over the ocean, and reached Fukuoka in the air mass.

(d) BC

To support the discussion of the domestic and the transboundary contributions to SNA, the behavior of BC at Fukuoka and the Goto Islands are shown in Fig. 8. The sensitivity simulation would suffer from a nonlinear chemistry response if complex atmospheric chemistry were involved; hence, we focused on BC, which is a primary aerosol. The temporal variation of BC also showed distinctive peaks for types N and S at Fukuoka and the Goto Islands. The model results reproduced these peaks

well, and the sensitivity simulation also suggested the dominance of transboundary air pollution for both peaks N and S. The temporal variation at the Goto Islands showed only two peaks of types N and S, although many short-term peaks were seen at Fukuoka. The sensitivity simulation confirmed that domestic air pollution contributed to these short-term peaks at Fukuoka; however, the model could not fully capture the peaks observed on January 7, 13, and 14. To improve the performance of the model to capture these short-term peaks, a higher-resolution model simulation and a revision of the





emission inventory are needed. Analysis of the primary aerosol confirmed that the transboundary air pollution was dominant for types N and S in January 2015.

Consequently, the well-validated model simulation indicated that two high $PM_{2.5}$ episodes with concentrations of around 100 $\mu g/m^3$ that occurred over western Japan during January were dominated by $NO_3^-$ for the first peak (type N) and by $SO_4^{2-}$ for

the second peak (type S), and that $NH_4^+$ concentration was high for both types. The model sensitivity simulation clarified that these high SNA concentrations in the type N and S episodes were dominated by the transboundary air pollution. In addition to the transport of SNA, abundant gas-phase $HNO_3$ and coarse-mode $NO_3^-$ reacted with sea-salt particles over the ocean and were also transported to western Japan. $NH_3$, which mainly came from domestic emissions, showed concentrations around zero during type N and S events, suggesting that $NH_3$ was depleted to neutralize $SO_4^{2-}$ and $NO_3^-$.

**3.2 Trajectory analysis**

Analyzing the synergetic observations at Fukuoka and the other three remote sites in Japan with the regional chemical transport model demonstrated that the two $PM_{2.5}$ episodic peaks were dominated by transboundary heavy pollution, even at Fukuoka. The two peaks had different SNA compositions. The first episode on January 11 showed a high $NO_3^-$ (type N) concentration and the second episode on January 17 was dominated by $SO_4^{2-}$ (type S). The differences in these episodes were

investigated further by a model simulation combined with backward trajectory analysis. The spatial distributions of $SO_4^{2-}$ and $NO_3^-$ during type N and S patterns are shown in Figs. 9 and 10, respectively.

In type N (Fig. 9 (right)), the model results showed that a low $SO_4^{2-}$ concentration of less than 5 $\mu g/m^3$ and a high $NO_3^-$ concentration of more than 10 $\mu g/m^3$ covered Fukuoka. The spatial distribution patterns indicated the outflow of $SO_4^{2-}$ and $NO_3^-$ from continental Asia to western Japan. The dominance of the transboundary air pollution suggested by these spatial

distributions was consistent with the model sensitivity simulation results (Figs. 5 and 6). High-concentration regions of $SO_4^{2-}$ and $NO_3^-$ stretched from the eastern coastline of China to the East China Sea and western Japan. The spatial distribution implied the direct transport from continental Asia to the downwind regions. In addition, the high-concentration region stretched from eastern China to western Japan, consistent with the corresponding $PM_{2.5}$ peak on January 11 at Shanghai and over Japan. To investigate the air mass origin for type N, the HYSPLIT backward trajectory (Stein et al., 2015) starting from

Fukuoka over 72 h was analyzed ($T_N$ in Fig. 9 (right)). The backward trajectory during type N transport suggested that the air mass originated from Shaanxi province and passed over Shanxi province, southern Hebei province, Shandong province, and then reached Fukuoka. The traveling time from the coast of China to Fukuoka was about 18 h. The distance from the coastline of China to Fukuoka is approximately 1000 km, so the air mass speed for type N was 55.6 km/h. Fig. 9 (left) shows the spatial distribution for when the air mass was located over China. A high concentration of $NO_3^-$ of more than 60 $\mu g/m^3$

occurred over the east coast of China before the air mass arrived in Fukuoka, whereas the $SO_4^{2-}$ concentration was as high as 10 $\mu g/m^3$ above the East China Sea.

In type S (Fig. 10 (right)), the model calculated that a high $SO_4^{2-}$ concentration of more than 20 $\mu g/m^3$ and a low $NO_3^-$ concentration of around 5 $\mu g/m^3$ covered Fukuoka. The HYSPLIT backward trajectory is shown as $T_S$ in Fig. 10. The air





mass during type S transport originated from Shanxi province and slowly moved over northern Henan province, Shandong province, and reached Fukuoka within about 24 h. The high-concentration regions stretched from eastern China to western Japan, consistent with the simultaneous $PM_{2.5}$ peak at Shanghai and over Japan. The air mass was stagnant over China compared with type N transport. The spatial distribution when the air mass was located over China is shown in Fig. 10 (left).

For $SO_4^{2-}$, the concentration was higher when the air mass arrived at Fukuoka compared with that in China, suggesting the fast production of $SO_4^{2-}$ during the transport process. A high concentration of $SO_4^{2-}$ of 20 μg/m$^3$ spread over the East China Sea and western Japan. A high $NO_3^-$ concentration of more than 60 μg/m$^3$ occurred over China, similar to type N; however, the $NO_3^-$ concentration was immediately reduced during the transport. The high $NO_3^-$ concentration of more than 10 μg/m$^3$ did not reach Fukuoka in this type S transport. Comparing the spatial distribution of the air mass over Fukuoka (Fig. 10

(right)) and China (Fig. 10 (left)) showed a 1 day delay in the high concentration peaks over Japan compared with the peak for Beijing (Figs. 3 and 4).

The backward trajectories for types N and S both showed similar transport patterns from China and Fukuoka. However, the spatial distribution patterns demonstrated clear differences between $SO_4^{2-}$ and $NO_3^-$ distributions for type N and S patterns. To discuss the different mechanisms for type N and S further, the model results were analyzed along the backward

trajectories of $T_N$ and $T_S$. The path analyses are shown in Figs. 11 and 12, along with additional indexes. The conversion ratio of SNA from the gas- to aerosol-phases is an important indicator. For $SO_4^{2-}$, considering the concentration of their gas-phase species of $SO_2$, the conversion ratio of Fs is defined as follows and calculated based on molar concentrations, shown in square brackets (Khoder, 2002).

$$F_s = \frac{[SO_4^{2-}]}{[SO_2] + [SO_4^{2-}]} \quad [mol/mol]. \quad (1)$$

A ratio of 0 indicates that $SO_4^{2-}$ was not produced, and a ratio of 1 indicates that $SO_2$ was converted completely to $SO_4^{2-}$. To confirm the Fs analysis, the concentration of the highly reactive hydroperoxy radical ($HO_2$) was also analyzed. Self-reaction of $HO_2$ produces hydrogen peroxide ($H_2O_2$), and this is the most effective oxidant of aqueous-phase $SO_2$ (Pandis and Seinfeld, 1989). Because the analyzed period was during winter, aqueous-phase reactions would be the main pathway for producing $SO_4^{2-}$.

For $NO_3^-$, the conversion ratios of Fn are defined by considering the concentration of $NO_2$ as a precursor and calculated based on molar concentrations (Khoder, 2002).

$$F_n = \frac{[HNO_3] + [NO_3^-]}{[NO_2] + [HNO_3] + [NO_3^-]} \quad [mol/mol]. \quad (2)$$

To investigate the ratio of $NO_3^-$ to gas- and aerosol-phase components, analogous to the calculation for $SO_4^{2-}$, the conversion ratio of Fn' is introduced.

$$F_n' = \frac{[NO_3^-]}{[HNO_3] + [NO_3^-]} \quad [mol/mol]. \quad (3)$$





The behavior of SNA is determined by introducing candidate indicators according to the work of Ansari and Pandis (1998) and Pinder et al. (2008). The gas ratio (GR) is an indicator of the sensitivity of $NO_3^-$ to changes in $SO_4^{2-}$ and $NH_4^+$ concentration. The GR is defined as the ratio of free ammonia to total nitrate as

$$GR = \frac{[NH_3+NH_4^+]-2[SO_4^{2-}]}{[HNO_3+ NO_3^-]} \quad [mol/mol], \quad (4)$$

where it is assumed that $SO_4^{2-}$ is fully neutralized, as indicated by the coefficient of 2 for $[SO_4^{2-}]$. The GR value implies the following condition for $NO_3^-$ production.

$$\begin{cases} GR > 1: NH_3 - rich \\ 0 < GR < 1: NH_3 - neutral \\ GR < 0: NH_3 - poor \end{cases} \quad (5)$$

Here, $NH_3$-rich means that there is sufficient $NH_3$ to neutralize both $SO_4^{2-}$ and $NO_3^-$, $NH_3$-neutral means that there is sufficient $NH_3$ to neutralize $SO_4^{2-}$, and $NH_3$-poor means that there is insufficient $NH_3$ to neutralize $SO_4^{2-}$ or $NO_3^-$. Free

ammonia, which can form $NH_4NO_3$ in the equilibrium process, is quantified more accurately by adjusting $[SO_4^{2-}]$ with the degree of sulfate neutralization (DSN). The DSN is defined as

$$DSN = \frac{[NH_4^+]-[NO_3^-]}{[SO_4^{2-}]} \quad [mol/mol], \quad (6)$$

where DSN is equal to or greater than 2 if there is sufficient $NH_4^+$. By substituting the coefficient of 2 before $[SO_4^{2-}]$ in Eq. (2) with Eq. (4), the adjusted GR (adjGR) can be defined as

$$adjGR = \frac{[NH_3+NH_4^+]-DSN\times[SO_4^{2-}]}{[HNO_3+ NO_3^-]} = \frac{[NH_3]+[NO_3^-]}{[HNO_3]+[NO_3^-]} \quad [mol/mol], \quad (7)$$

by using the indexes, and the path analysis of backward trajectories for the model results are shown in Figs. 11 and 12, and are summarized in Table 1. The analyses are shown for (a) the meteorological components of temperature and relative humidity, (b) $SO_4^{2-}$ and $SO_2$ with an enlarged view for $SO_4^{2-}$, (c) $HO_2$ concentration, (d) $NO_3^-$, $HNO_3$, NO, $NO_2$, other $NO_y$ consisting of $NO_3$, $HNO_2$, $N_2O_5$, and peroxyacyl nitrates (PANs), and coarse-mode $NO_3^-$, (e) $NH_4^+$ and $NH_3$, and (f) adjGR,

Fs, and Fn'. The concentrations of the air pollutants were reduced by chemical reactions, dispersion, and deposition during the transport; therefore, as an index of dispersion and deposition processes, BC and CO concentrations normalized to their maximum concentrations during the transport were used in the variation of the total concentration in (b), (d), and (e). Table 1 shows the meteorological components and the air pollutant concentrations of each component and total sulfate, total nitrate, and total ammonia. The indexes were averaged over China, the transport time above the ocean, and Fukuoka. The SNA

concentration was balanced between $SO_4^{2-}$ with $NO_3^-$ and $NH_4^+$ during both types N and S.

During the type N pattern (Fig. 11), the concentration of air pollutants increased after the air mass moved into Hebei province (-30 h). The relative humidity was lower than 40% when the air mass traveled over Shanxi and Hebei provinces (Fig. 11a). The $SO_4^{2-}$ concentration was around 1 μg-S/m³ during transport (Fig. 11b and Table 1), and $SO_2$ was dominant in



the total sulfate concentration. In this type N pattern, the concentration of the most effective oxidant of $HO_2$ for the $SO_2$ aqueous-phase reaction was smaller due to the lower relative humidity (Fig. 11c), and the conversion ratio was less than 0.1 (Fig. 11f and Table 1). Compared with the $SO_4^{2-}$ variation, as the trajectory passed from Shanxi province (-36 h) to Shandong province (-18 h), and over the ocean (-12 h), $NO_3^-$ and $NH_4^+$ concentration increased. The concentration of $NO_3^-$

and $NH_4^+$ decreased as the air mass traveled to Japan (Figs. 11d and 11e). Over China (-30 to -18 h), gas-phase $NH_3$ was abundant and $HNO_3$ was fully consumed to produce $NH_4NO_3$ (Figs. 11d and 11e). The excess $HNO_3$ over the ocean contributed to producing coarse-mode $NO_3^-$ by reacting with sea-salt particles as the air mass traveled over the ocean (Fig. 11d). The adjGR was super $NH_3$-rich over China (Fig. 11f). After the air mass left Shandong, $HNO_3$ increased and $NH_3$ was close to zero (Figs. 11d and 11e), so adjGR shifted to $NH_3$-neutral status (Fig. 11f). Consequently, the proportion of $NO_3^-$,

indicated by Fn', remained around 80% during transport from China to Japan (Fig. 11f). The rate of the decrease of total sulfate, total nitrate, and total ammonia were generally consistent with the rate of decrease of normalized BC and CO, suggesting that the budget was almost satisfied during transport (Table 1). The decrease of BC was larger than that of CO because of wet deposition (e.g., Pan et al., 2011, Kanaya et al., 2016).

During the type S pattern (Fig. 12), SNA concentrations were higher compared with the type N pattern (Fig. 11), partly due

to the slower motion of the air mass compared with type N. $SO_4^{2-}$ concentration was around 2 μg-S/m$^3$ over Shanxi to Henan provinces (-72 to -36 h), it increased slightly to 3 μg-S/m$^3$ over Henan to Shandong provinces (-36 to -24 h), and subsequently increased to 3 μg-S/m$^3$ over the ocean (Fig. 12b and Table 1). The aqueous-phase reaction contributed to the production, as suggested by the variation in $HO_2$ concentration. Over oceans, the relative humidity was around 80%, 20% larger than in the case of the type N pattern (Fig. 12a and Table 1). The Fs conversion ratio was 0.3 when the air mass

reached Fukuoka (Fig. 12f). The increase in $SO_4^{2-}$ over Fukuoka compared with over China was +67.2% (Table 1). The behaviors of gas-phase $HNO_3$ and $NH_3$ were similar in type N patterns (Figs. 12d and 12e); the $NH_3$ concentration was high over China, and $HNO_3$ gradually increased after the air mass approached the ocean, partly because coarse-mode $NO_3^-$ was produced. However, the $NH_3$ concentration was smaller compared with type N patterns and remained near zero when the air mass passed over Henan province (Fig. 12e). adjGR showed slight $NH_3$-rich status over China (-54 to -36 h), and shifted to

$NH_3$-neutral status before the air mass left China (-36 h), and at the same time, the ratio of $NO_3^-$ indicated by Fn' decreased to 0.45 (Fig. 12f and Table 1). Similar to type N, the rates of decrease of total sulfate, total nitrate, and total ammonia were generally consistent with the rates of decrease of normalized BC and CO.

To summarize the key points for type N and S patterns, the behavior of $SO_4^{2-}$, $NO_3^-$, and $HNO_3$ concentrations during transport from China to Japan are shown in Fig. 13. The relative percentages of the concentration of each species to the total

concentration of $SO_4^{2-}$, $NO_3^-$, and $HNO_3$ are shown. Types N and S both showed the dominance of $NH_4NO_3$ (over 70%) when the air mass was over China (-48 h). For the air mass close to Fukuoka, because there was no $NH_3$, $NH_4NO_3$ decomposed into gas-phase $NH_3$ and $HNO_3$. Therefore, $HNO_3$ concentration increased as the air mass reached Fukuoka for types N and S. At this time, the $SO_4^{2-}$ concentration was important in determining the $NO_3^-$ concentration. $SO_4^{2-}$ production (Figs. 11b and 12b) through an aqueous-phase reaction was indicated by the $HO_2$ concentration (Figs. 11c and 12c), and Fs



showed a large difference between types N (lower than 0.1 during transport; Figs. 11f) and S (around 0.3 at Fukuoka; Figs. 12f). Once $H_2SO_4$ was produced via an aqueous-phase reaction, it reacted with gas-phase $NH_3$ to produce $(NH_4)_2SO_4$, leading to further decomposition of $NH_4NO_3$. The ratio of $NO_3^-$ (Fn') showed different behavior for types N (around 0.8 during transport; Figs. 11f) and S (lower than 0.5 at Fukuoka; Figs. 12f). The transport pattern under similar conditions over

China was determined by the low $SO_4^{2-}$ concentration maintaining a higher $NO_3^-$ concentration at Fukuoka (type N), or by $SO_4^{2-}$ production under a higher relative humidity resulting in the dominance of $SO_4^{2-}$ with further $NH_4NO_3$ decomposition (type S).

Finally, the outflow of $SO_4^{2-}$ and $NO_3^-$ from China to western Japan during the intensive observation campaign is summarized in Fig. 14. In this figure, the modeled $SO_4^{2-}$ and $NO_3^-$ concentrations were averaged at 32–36°N to cover the

four sites in Japan (Fig. 3) and are shown as a time-longitude cross section. The longitude of Shanghai, the Goto Islands, Tsushima Island, Fukuoka, and Tottori are indicated at the bottom of the figure. This outflow analysis can help to identify the areas affected by transboundary heavy pollution. The main outflow from China to western Japan occurred twice (types N and S). The outflow concentration of $SO_4^{2-}$ was lower on January 11 for type N and larger on January 17 for type S; a high concentration of more than 15 μg/m$^3$ reached 130°N (Fukuoka) and a concentration of around 10 μg/m$^3$ reached 134°N

(Tottori) for type S (Fig. 14a). The outflow of $NO_3^-$ was observed over the Goto Islands, Tsushima Island, and Fukuoka on January 11 for type N, whereas the high concentration of over 10 μg/m$^3$ was limited to the East China Sea regions on January 17 for type S. A concentration of $NO_3^-$ of more than 5 μg/m$^3$ did not reach 134°N (Tottori) for type N (Fig. 14b). The outflow analysis suggested that $SO_4^{2-}$ can be transported longer distances, whereas transboundary air pollution of $NO_3^-$ is limited to western Japan, especially over Kyushu.

**4 Conclusion**

Based on the state-of-the-art observation systems for capturing SNA behavior and the chemical transport model, two episodes of high $PM_{2.5}$ concentrations of around 100 μg/m$^3$ occurred during winter over western Japan were analyzed. The first episode on January 11 was dominated by $NO_3^-$ (type N) and the second episode on January 17 was dominated by $SO_4^{2-}$ (type S). The chemical transport model captured the behavior of SNA and the related gas-phase species of $HNO_3$ and $NH_3$,

and coarse-mode $NO_3^-$ observed over Japan. The model also reproduced $PM_{2.5}$ variation over China. To evaluate the domestic contributions, sensitivity analysis was performed, in which the anthropogenic emissions from Japan were switched off in the chemical transport model. The results showed that there were sometimes domestic contributions for $NO_3^-$, although the type N and S patterns were dominated by the transboundary air pollution, even at Fukuoka. The effect of transboundary air pollution on type N and S patterns were also confirmed by analyzing the behavior of BC at Fukuoka and the remote Goto

Islands. The importance of the transboundary air pollution for coarse-mode $NO_3^-$, produced by abundant $HNO_3$ and sea-salt particles, was also revealed. To investigate the characteristic differences between type N and S patterns, the chemical transport model results were analyzed by the backward trajectory analysis from Fukuoka to continental Asia. We also





evaluated the adjusted gas ratio (adjGR), which indicates the sensitivity of $NO_3^-$ to changes in $SO_4^{2-}$ and $NH_4^+$, and Fs, which is the conversion ratio of $SO_2$ to $SO_4^{2-}$. For the $SO_2$ aqueous-phase reaction, $H_2O_2$ is the most effective oxidant. Thus, $HO_2$, which produces $H_2O_2$ through self-reaction, was also analyzed. The features of type N and S patterns are summarized as follows.

● In the type N transport pattern, $NO_3^-$ was mainly the $NH_4^+$ counterion during transport from China to Japan. A high $NO_3^-$ concentration of more than 10 μg/m³ was observed at Fukuoka. $NH_3$ was abundant and $HNO_3$ was completely consumed to produce $NO_3^-$ over China; hence adjGR indicated super $NH_3$-rich status, which meant full neutralization of both $SO_4^{2-}$ and $NO_3^-$, over China. After the air mass left China, $HNO_3$ increased and $NH_3$ was close to zero; so adjGR shifted to $NH_3$-neutral status. The $SO_4^{2-}$ concentration was always lower than the $NO_3^-$ concentration. This was
because the $HO_2$ concentration was less than 3 pptv during transport from China to Fukuoka, and Fs suggested a slow conversion ratio for $SO_4^{2-}$ of less 0.1. $SO_4^{2-}$ production via an aqueous-phase reaction was also inhibited. This also explained why the air mass maintained a higher $NO_3^-$ concentration during transport.

● In the type S transport pattern, the ion balance between $NH_4^+$ and $NO_3^-$ with $SO_4^{2-}$ showed that the counterion of $NH_4^+$ was mainly $NO_3^-$ over China, and then became $SO_4^{2-}$ as the air mass left China and approached Fukuoka. A high
concentration of $SO_4^{2-}$ of more than 20 μg/m³ was observed at Fukuoka. The change in Fs from 0.1 to 0.3 when the air mass reached Fukuoka was consistent with this observation. Higher Fs values were related to higher relative humidity and $HO_2$ concentration, indicating the high production of $SO_4^{2-}$ via an aqueous-phase reaction ($H_2SO_4$). The production of $H_2SO_4$ promoted the reaction with $NH_3$ to produce $(NH_4)_2SO_4$ and further decomposition of $NH_4NO_3$ during the transport process from China to Fukuoka. The temporal behaviors of gas-phase $HNO_3$ and $NH_3$ were similar for type
N; however, the $NH_3$ concentration was lower. adjGR showed almost $NH_3$-neutral conditions during type S. The production of $SO_4^{2-}$ and the insufficient supply of $NH_3$ contributed to the rapid decomposition of $NH_4NO_3$ in this case.

In this study, we clarified the two types transport pattern for SNA. The spatial distribution pattern of the outflow over East Asia during January 2015 showed that the outflow of $SO_4^{2-}$ stretched over the whole of western Japan, whereas the transboundary air pollution of $NO_3^-$ played an important role over Kyushu Island, western Japan. Generally, the
transboundary air pollution dominated by $SO_4^{2-}$ (type S) has been recognized over East Asia, but we have elucidated the effect of the transboundary heavy pollution dominated by $NO_3^-$ (type N). Our findings will promote the importance of $NO_3^-$ long-range transport.

The study period was limited to January 2015, so the analyzed period should be extended to investigate the type S and N transport patterns further. The variation of the gas ratio for emissions (GRe), which considers the balance between $SO_2$, NOx,
and $NH_3$ (analogous to Eq. 4) may be useful for future analysis. The emission reductions achieved by the 12th Chinese Five Year Plan during 2011–2015 (Asia Society, 2016), resulted in a GRe increase, which caused an $NH_3$-rich status compared with the current status. Moreover, the target for the 13th Chinese Five Year Plan during 2016–2020, suggests there will be further increases in GRe. The effects of different reduction rates for SNA precursor gases on transboundary air pollution of $NO_3^-$, especially over western Japan, should be modeled.





**Author contributions**

I. Uno designed the synergetic observations at Chikushi Campus of Kyushu University and other remote sites in western Japan. S. Yamamoto and K. Osada respectively carried out the ground-based ACSA and NHx-monitor observations at Fukuoka. Y. Kamiguchi and K. Osadaconducted air sampling and chemical analysis for D-F pack samples during the
5 intensive observation period at Fukuoka. K. Osada and Y. Kurosaki measured $PM_{2.5}$ and analyzed samples at Tottori. K. Tamura analyzed the observations from the remote sites of Tsushima Island and the Goto Islands. Y. Kanaya conducted the BC observations at Fukuoka and the Goto Islands. S. Itahashi developed the modeling system, performed the model simulations and analysis, and prepared the manuscript with contributions from all co-authors.

**Acknowledgements**

10 This work was partly supported by Japan Society for the Promotion of Science (JSPS) KAKENHI Grant Numbers JP25220101, JP23310004, JP15H02803. This work was also partly supported by the Environment Research and Technology Development Fund of the Ministry of the Environment, Japan (No. S-7, 2-1403, 5-1505). This work was partly funded by the Joint Research Program of Arid Land Research Center, Tottori University (No. 28C2011) and by the Collaborative Research Program of Research Institute for Applied Mechanics, Kyushu University (No. 26AO-2, 27AO-6, 28AO-2). We thank Dr.
15 Sayako Ueda at Nagoya University for analyzing tape filters at Tottori University.




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

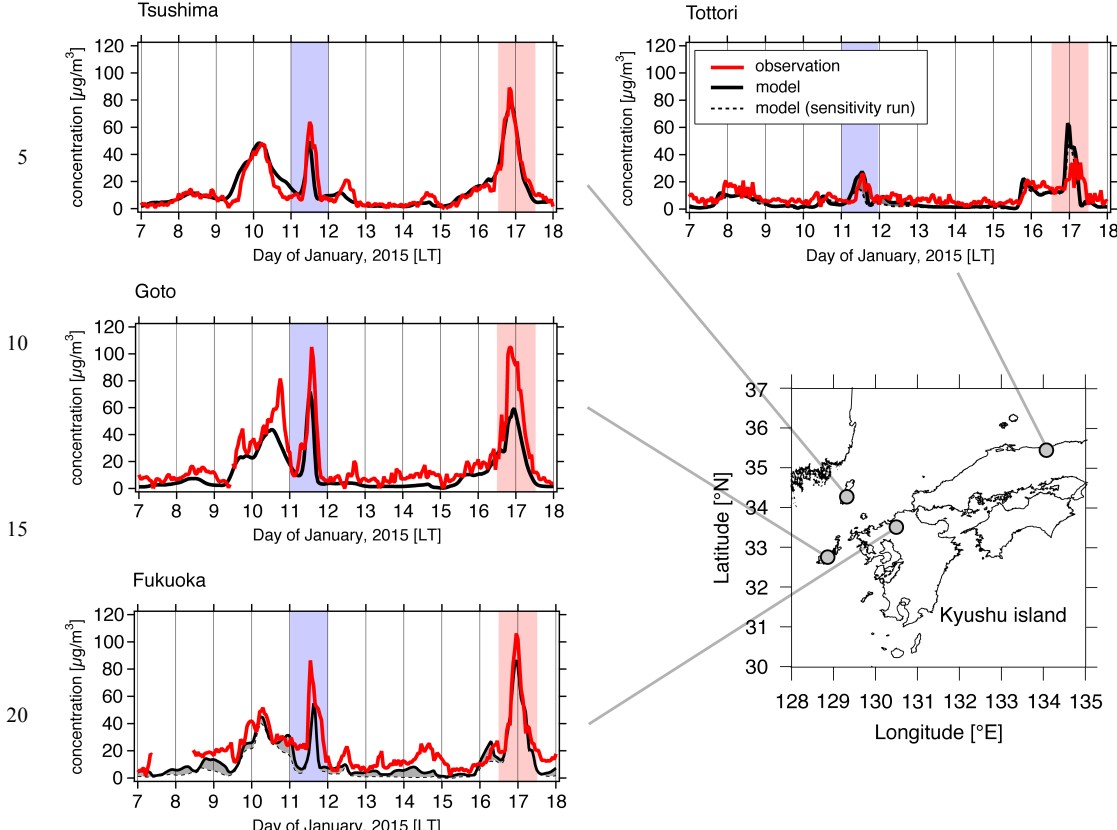

Figure 1: Temporal variation of PM$_{2.5}$ over Japan at Fukuoka, the Goto Islands, Tsushima Island, and Tottori during January 7-17, 2015. Blue and red shading show the episodes focused on in this study. Red lines indicate observations. Black lines indicate the base case simulation and dotted black lines indicate the sensitivity simulation in which the anthropogenic emissions from Japan were switched off; the differences between these results shown in gray represent local contributions.




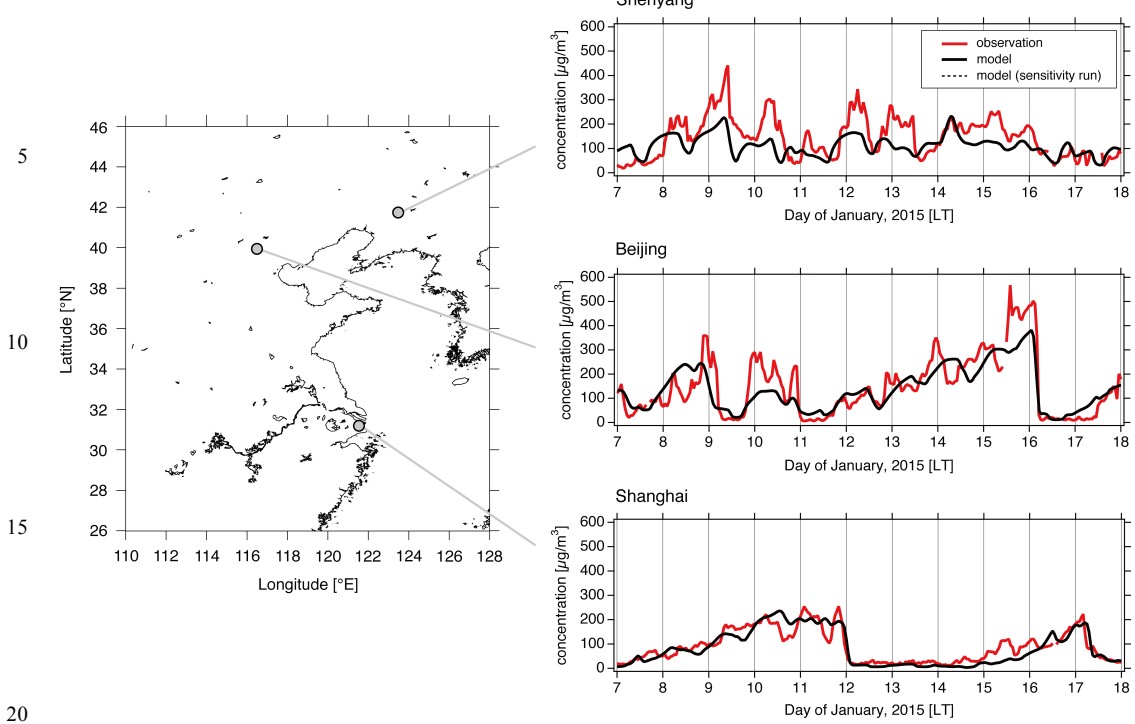

**Figure 2: Temporal variation of PM$_{2.5}$ over China at Beijing, Shanghai, and Shenyang during January 7-17, 2015. Red lines indicate observations by BAM at the U.S. Embassy in Beijing and at the U.S. Consulates in Shenyang and Shanghai. Black lines indicate the base case simulation and dotted black lines indicate the sensitivity simulation in which the anthropogenic emissions from Japan were switched off; the differences between these results shown in gray represent Japanese contributions.**



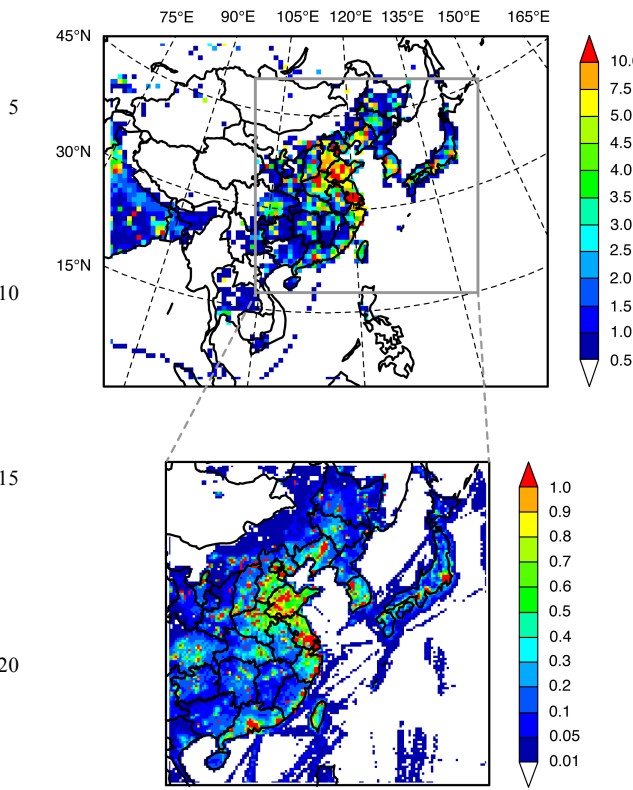

**Figure 3: Modeling domain for horizontal resolutions of (top) 81 km and (bottom) 27 km with anthropogenic NOx emissions.**



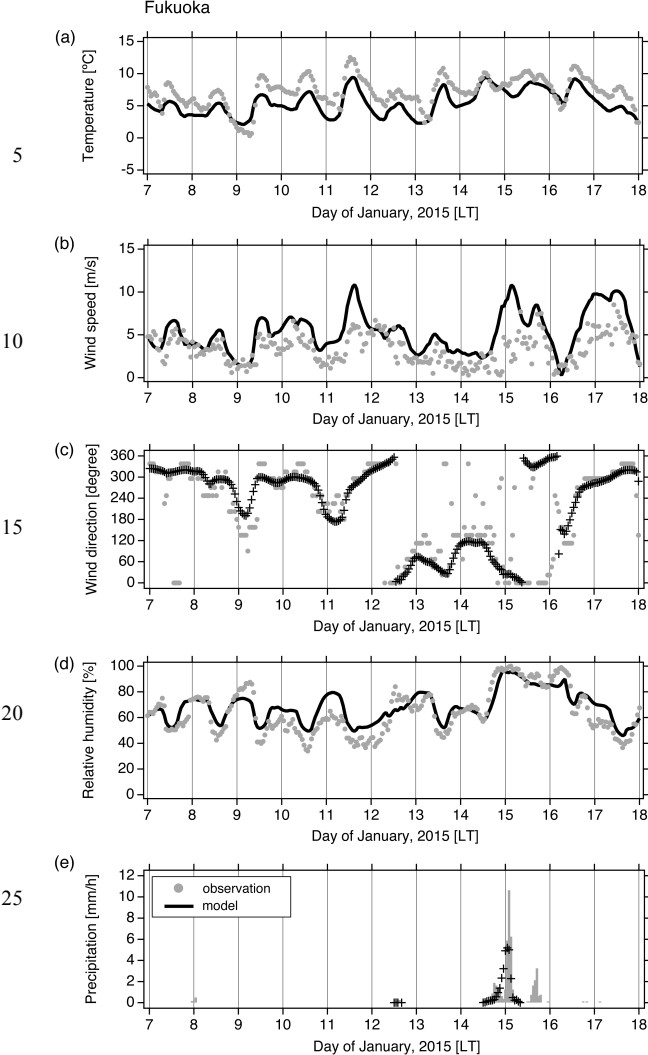

**Figure 4: Temporal variation of (a) temperature, (b) wind speed, (c) wind direction, (d) relative humidity, and (e) precipitation at Fukuoka during January 7-17, 2015. Gray and black indicate observations and model results, respectively.**



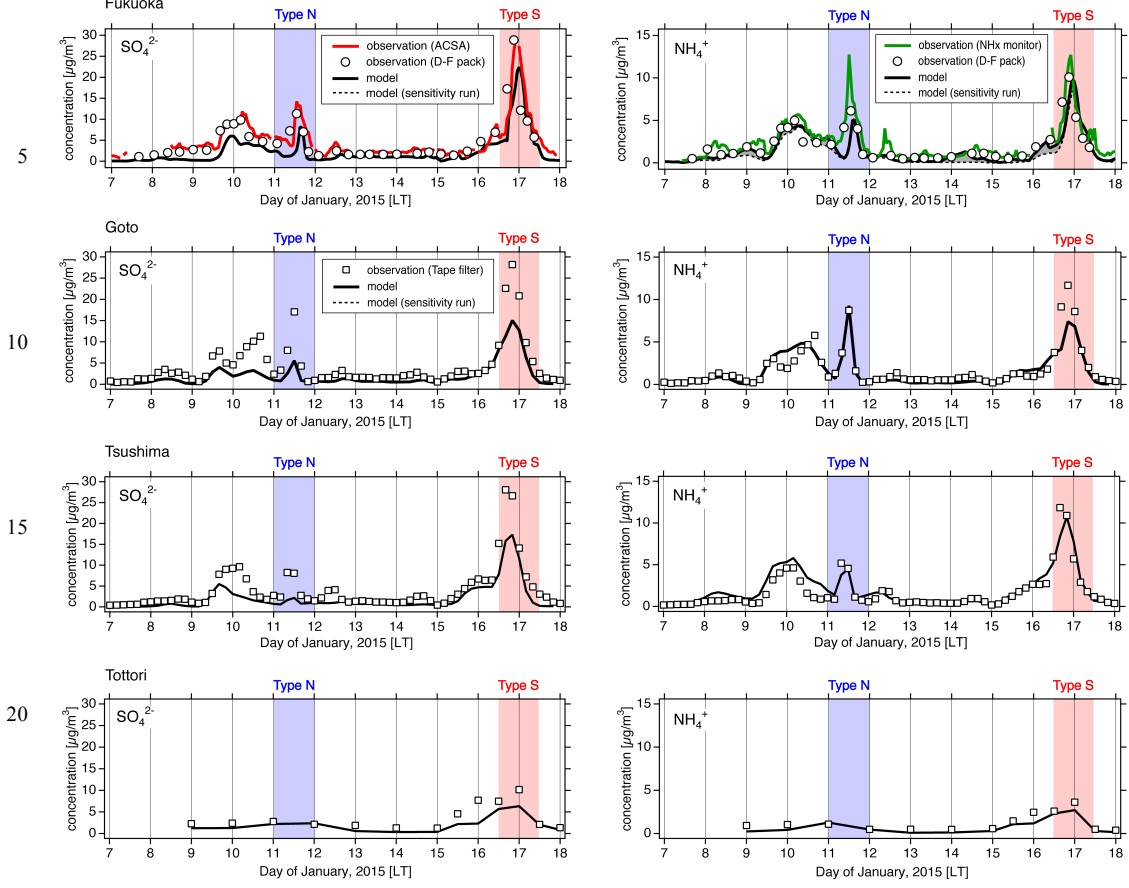

**Figure 5: Temporal variation of SO$_4^{2-}$ and NH$_4^+$ over Japan at Fukuoka, the Goto Islands, Tsushima Island, and Tottori during January 7-17, 2015. Blue and red shading show the type N and S patterns focused on in this study. Red lines indicate SO$_4^{2-}$ observations by ACSA and green lines indicate NH$_4^+$ observations by NHx monitor at Fukuoka. Open circles are D-F pack observations at Fukuoka. Open squares are tape filter measurements at the Goto Islands, Tsushima Island, and Tottori. Black lines indicate the base case simulation and dotted black lines indicate the sensitivity simulation in which the anthropogenic emissions from Japan were switched off; the differences between these results shown in gray represent local contributions.**



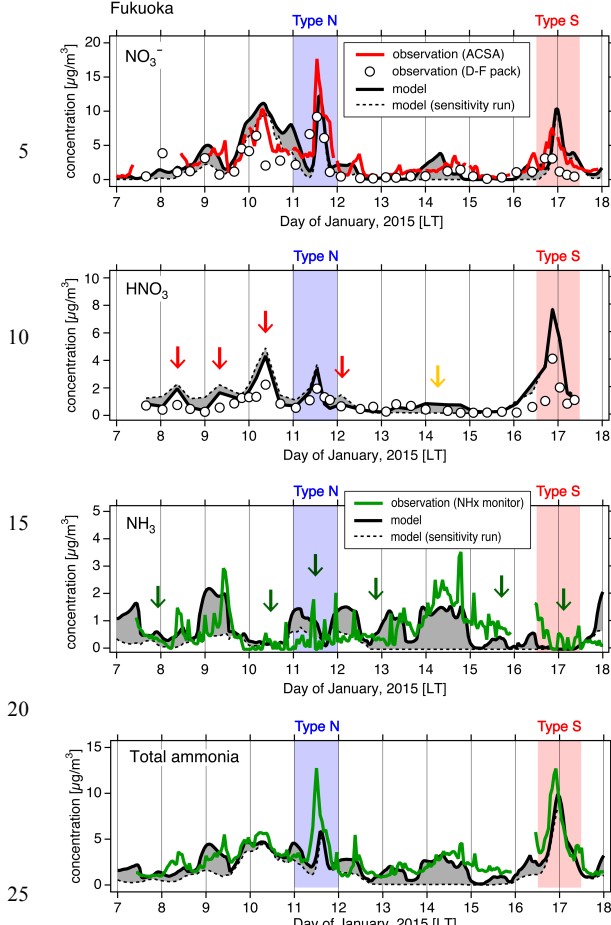

**Figure 6: Temporal variation of NO₃⁻, HNO₃, NH₃, and total ammonia at Fukuoka during January 7-17, 2015. Blue and red shading show the type N and S patterns focused on in this study. Red lines indicate $NO_3^-$ observations by ACSA, green lines indicate $NH_3$ and total ammonia observations by NHx monitor, and open circles indicate D-F pack observations. For $NH_3$, periods of nearly zero concentration (24 h average of less than 1 μg/m³) are indicated by arrows. Black lines indicate the base case simulation and dotted black lines indicate the sensitivity simulation in which the anthropogenic emissions from Japan were switched off; the differences between these results shown in gray represent local contributions.**



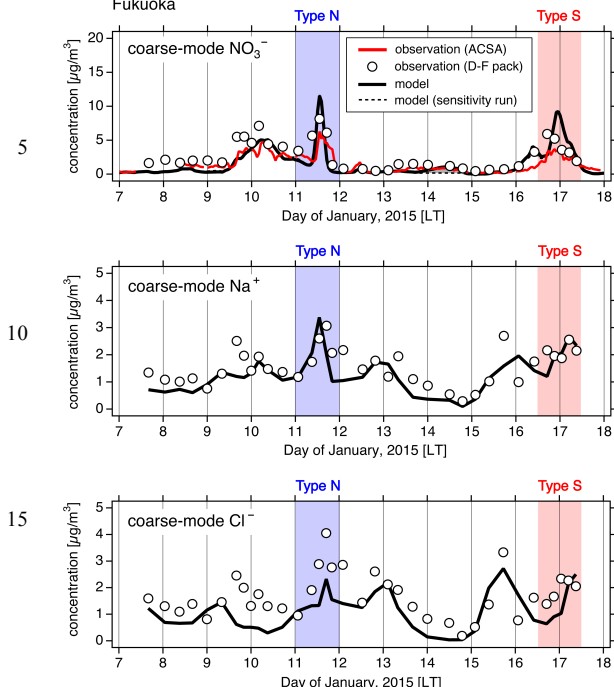

**Figure 7: Temporal variation of coarse-mode NO$_3^-$, Na$^+$, and Cl$^-$ at Fukuoka during January 7-17, 2015. Blue and red shading show the type N and S patterns focused on in this study. Red lines indicate coarse-mode NO$_3^-$ observations by ACSA and open circles indicate D-F pack observation. Black lines indicate the base case simulation and dotted black lines indicate the sensitivity simulation in which the anthropogenic emissions from Japan were switched off; the differences between these results shown in gray represent local contributions.**

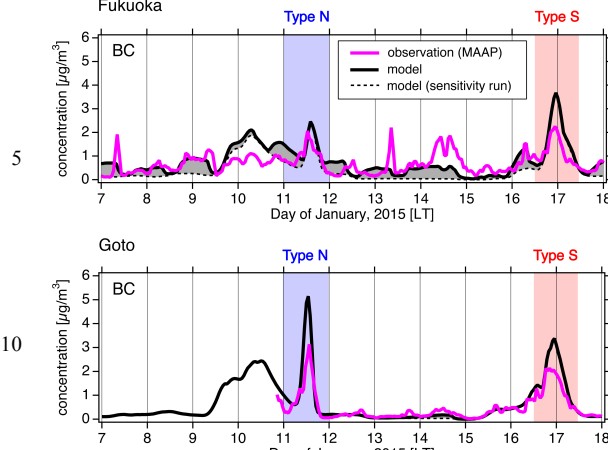

**Figure 8: Temporal variation of BC at Fukuoka and the Goto Islands during January 7-17, 2015. Blue and red shading show the type N and S patterns focused on in this study. The magenta line indicates BC observations by MAAP. Black lines indicate the base case simulation and dotted black lines indicate the sensitivity simulation in which the anthropogenic emissions from Japan were switched off; the differences between these results shown in gray represent local contributions.**



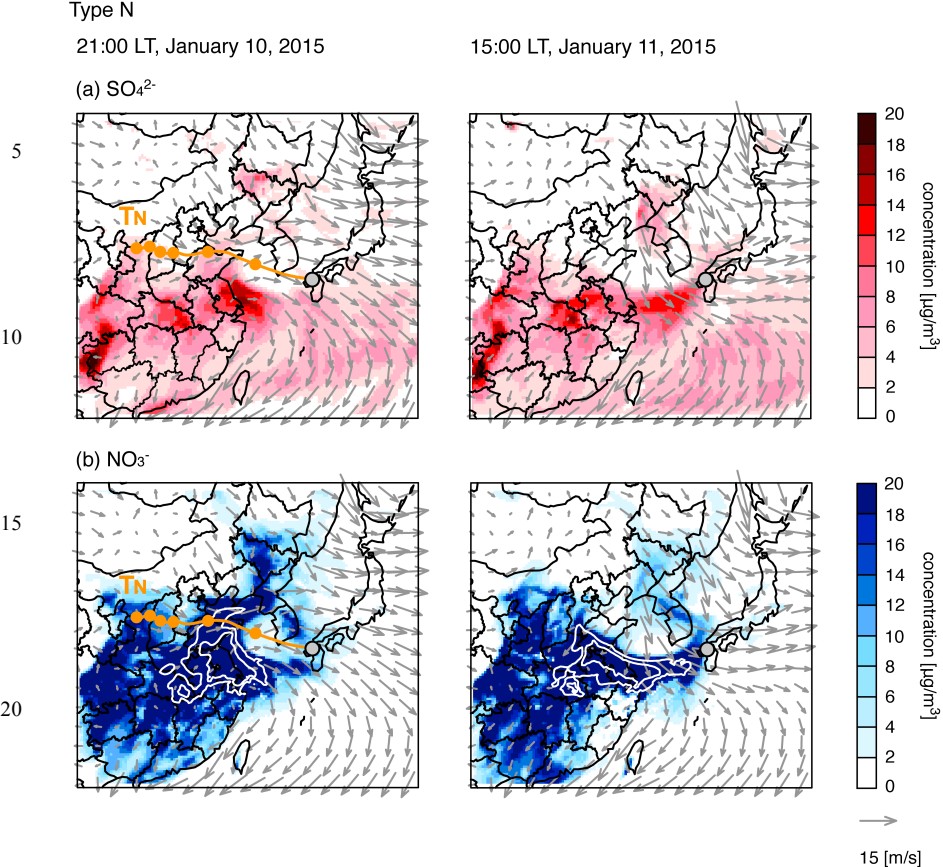

**Figure 9: Simulated spatial distribution for (a) SO$_4^{2-}$ and (b) NO$_3^-$ during type N pattern. Contours shown by white lines for NO$_3^-$ represent 40 and 60 μg/m$^3$. The 72 h HYSPLIT backward trajectory from Fukuoka is overlaid with orange lines with circles at 12 h intervals. (Left) 21:00 LT, January 10, 2015, when the air mass left China, and (right) 15:00 LT, January 11, 2015, when the air mass reached Fukuoka.**




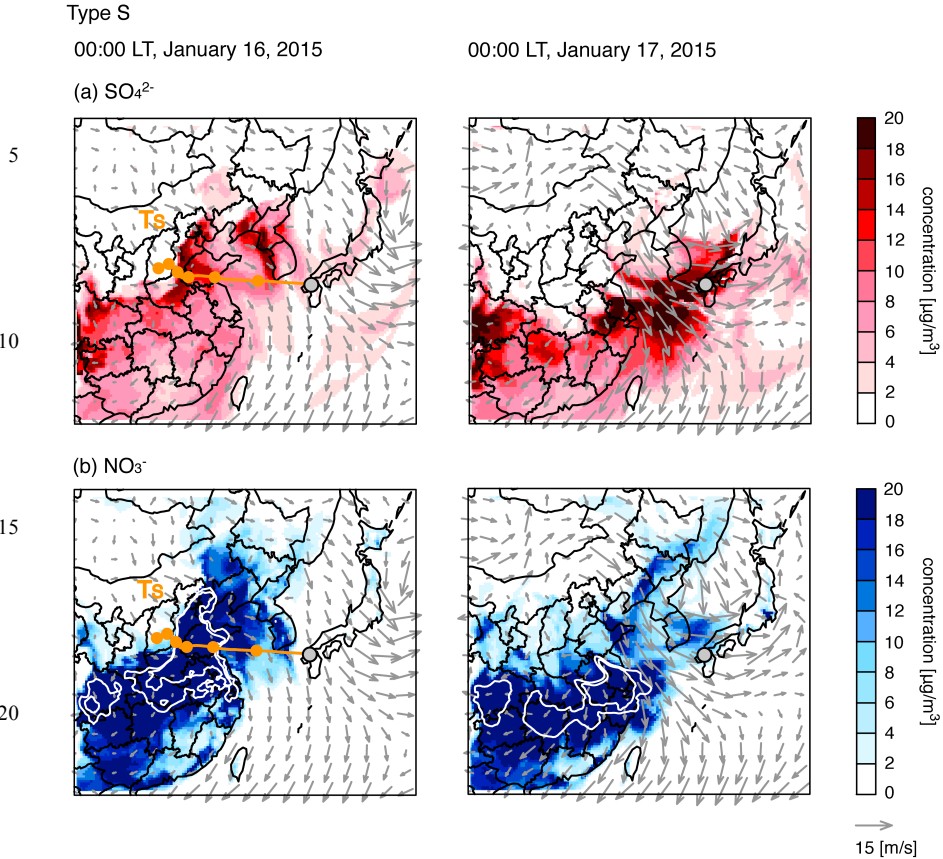

**Figure 10: Simulated spatial distribution for (a) SO$_4^{2-}$ and (b) NO$_3^-$ during type S pattern. Contours shown by white lines for NO$_3^-$ represent 40 and 60 μg/m$^3$. The 72 h HYSPLIT backward trajectory from Fukuoka is overlaid with orange lines with circles at 12 h intervals. (Left) 00:00 LT, January 16, 2015, when the air mass left China, and (right) 00:00 LT, January 17, 2015, 00LT when the air mass reached Fukuoka.**





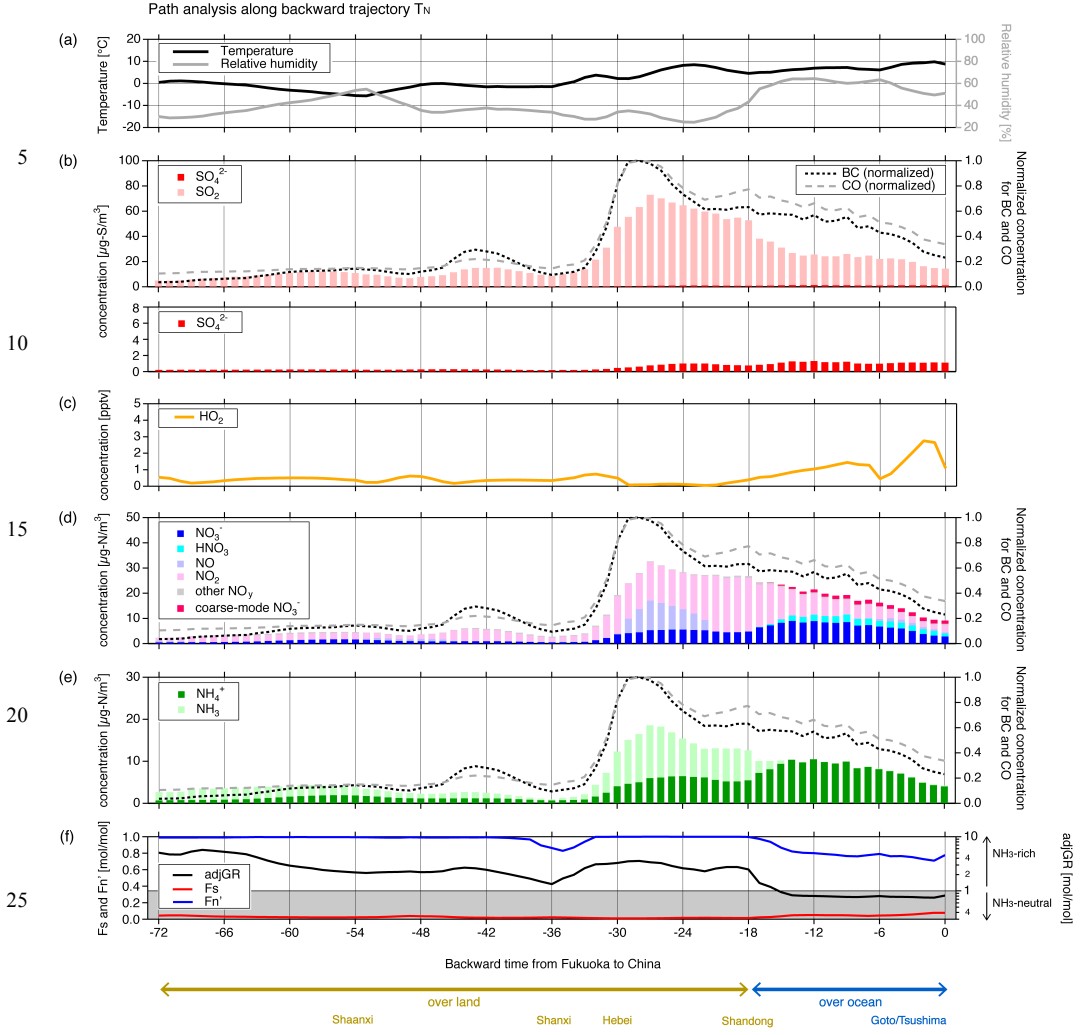

**Figure 11: Path analysis of model results along trajectory T$_N$. (a)** Temperature and relative humidity, **(b)** SO$_4^{2-}$ (with expansion at the bottom) and SO$_2$, **(c)** HO$_2$ concentration, **(d)** NO$_3^-$, HNO$_3$, NO, NO$_2$, other NO$_y$ (NO$_3$, HNO$_2$, N$_2$O$_5$, and PANs), and coarse-mode NO$_3^-$, **(e)** NH$_4^+$ and NH$_3$, **(f)** adjGR, Fs, and Fn'. In (b), (d), and (e), BC and CO concentrations normalized to the maximum value are also shown. Time axis indicates the backward time from Fukuoka. Brown and blue bars at the bottom are schematic images of the trajectory location over land and ocean.





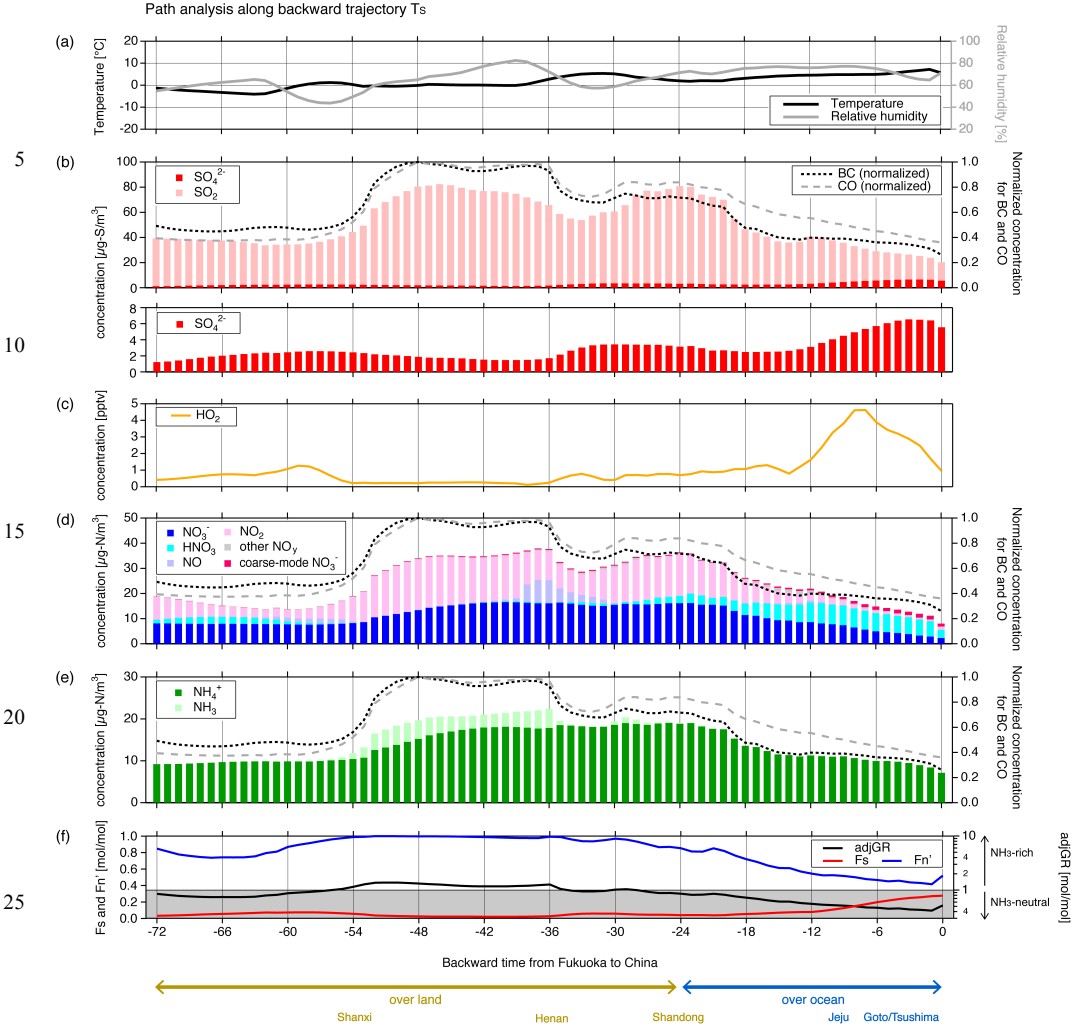

**Figure 12: Path analysis of model results along trajectory $T_S$. (a) Temperature and relative humidity, (b) $SO_4^{2-}$ (with expansion at the bottom) and $SO_2$, (c) $HO_2$ concentration, (d) $NO_3^-$, $HNO_3$, $NO$, $NO_2$, other $NO_y$ ($NO_3$, $HNO_2$, $N_2O_5$, and PANs), and coarse-mode $NO_3^-$, (e) $NH_4^+$ and $NH_3$, (f) adjGR, Fs, and Fn'. In (b), (d), and (e), BC and CO concentrations normalized to the maximum value are also shown. Time axis indicates the backward time from Fukuoka. Brown and blue bars at the bottom are schematics of the trajectory location over land and ocean.**



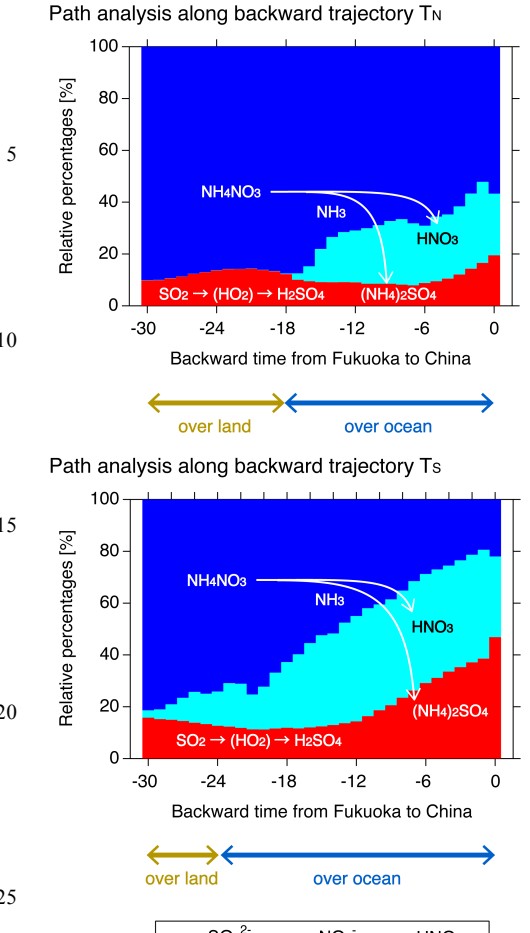

**Figure 13: Path analysis of model results along trajectories $T_N$ and $T_S$. Percentages of each concentration relative to the sum of the $SO_4^{2-}$, $NO_3^-$, and $HNO_3$ concentrations in equivalent units are shown. Time axis indicates the backward time from Fukuoka. Brown and blue bars at the bottom are schematics of the trajectory location over land and ocean.**



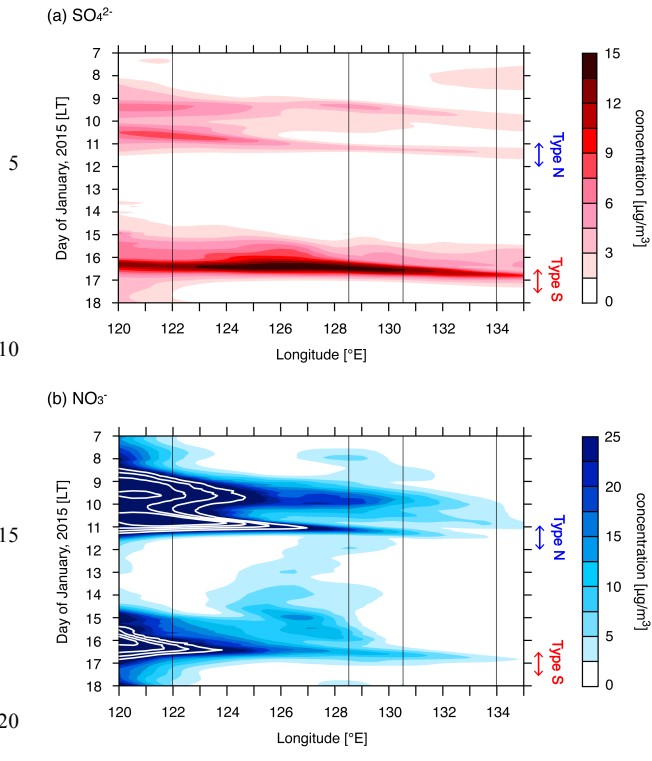

**Figure 14: Outflow frequency of (a) SO₄²⁻ and (b) NO₃⁻. Contours shown by white lines for NO₃⁻ represent 30, 40, 50, and 60 µg/m³. Model results are averaged over 32–36°N, and are shown as the time-longitude cross section. Longitudes of representative locations are indicated by thin black lines and the location names are written in brown at the bottom of the figure.**



Table 1. Summary of path analysis for types N and S.

| Type | Type N | | | Type S | | |
|---|---|---|---|---|---|---|
| Arrival time at Fukuoka | 15:00 LT, January 11 | | | 00:00 LT, January 17 | | |
| Transport time from China | 18 h | | | 24 h | | |
| | Over China | During transport | Over Fukuoka | Over China | During transport | Over Fukuoka |
| Temperature [°C] | 6.6 | 6.7 | 9.3 | 3.1 | 4.0 | 6.5 |
| Relative humidity [%] | 32.7 | 60.5 | 50.5 | 66.3 | 74.4 | 67.0 |
| $SO_4^{2-}$ [µg-S/m$^3$] | 0.9 | 1.1 | 1.1 | 3.3 | 3.7 | 6.2 |
| | | (+24.5%) | (+25.5%) | | (+14.1%) | (+67.2%) |
| $SO_2$ [µg-S/m$^3$] | 55.9 | 25.0 | 14.1 | 72.1 | 40.0 | 17.0 |
| Total sulfate [µg-S/m$^3$] | 56.8 | 26.1 | 15.2 | 75.4 | 43.8 | 23.1 |
| | | (-54.1%) | (-74.7%) | | (-41.9%) | (-73.2%) |
| Fs [mol/mol] | 0.02 | 0.04 | 0.07 | 0.04 | 0.10 | 0.27 |
| $NO_3^-$ [µg-N/m$^3$] | 4.9 | 7.5 | 3.3 | 15.9 | 9.4 | 2.8 |
| | | (+53.6%) | (-42.0%) | | (-41.1%) | (-85.6%) |
| $HNO_3$ [µg-N/m$^3$] | 0.0 | 2.1 | 1.7 | 1.9 | 6.1 | 5.2 |
| Other NOy [µg-N/m$^3$] | 22.2 | 8.5 | 3.3 | 16.7 | 6.4 | 0.9 |
| Coarse-mode $NO_3^-$ [µg-N/m$^3$] | 0.0 | 1.1 | 1.4 | 0.4 | 0.7 | 1.4 |
| Total NOy [µg-N/m$^3$] | 27.1 | 19.1 | 9.7 | 34.8 | 22.5 | 10.4 |
| | | (-29.3%) | (-66.2%) | | (-35.3%) | (-77.0%) |
| Fn' [mol/mol] | 1.00 | 0.81 | 0.73 | 0.90 | 0.61 | 0.45 |
| $NH_4^+$ [µg-N/m$^3$] | 5.6 | 8.7 | 4.4 | 18.8 | 12.6 | 8.2 |
| | | (+54.3%) | (-28.9%) | | (-33.0%) | (-62.1%) |
| $NH_3$ [µg-N/m$^3$] | 7.5 | 0.4 | 0.2 | 0.6 | 0.1 | 0.0 |
| Total ammonia [µg-N/m$^3$] | 13.2 | 9.1 | 4.6 | 19.5 | 12.7 | 8.2 |
| | | (-30.7%) | (-67.9%) | | (-34.8%) | (-63.3%) |
| adjGR [mol/mol] | 2.56 | 0.87 | 0.77 | 0.93 | 0.61 | 0.46 |
| BC [µg/m$^3$] | 6.5 | 5.2 | 2.6 | 9.7 | 6.1 | 4.0 |
| | | (-20.2%) | (-63.3%) | | (-37.4%) | (-63.9%) |
| CO [ppbv] | 904.4 | 737.6 | 440.2 | 1260.0 | 887.0 | 564.9 |
| | | (-18.4%) | (-53.8%) | | (-29.8%) | (-56.8%) |
| Key points | • Dominance of $NO_3^-$ compared with $SO_4^{2-}$ <br> • Under 10% conversion ratio for $SO_4^{2-}$ <br> • Lower relative humidity <br> • Abundant $NH_3$ supply above China <br> • $NH_3$-rich air mass maintaining neutralization of $NO_3^-$ | | | • Dominance of $SO_4^{2-}$ compared with $NO_3^-$ <br> • Approximately 30% conversion ratio for $SO_4^{2-}$ <br> • Higher relative humidity of around 70% <br> • No $NH_3$ in gas-phase <br> • $NH_3$-neutral conditions during transport, $SO_4^{2-}$ neutralized | | |

Note: Parentheses indicate the multiplying factors compared with the status over China. The status is averaged over 6 h before the air mass leaves China, over during transport time from China to Fukuoka above the ocean, and over 3 h before the air mass reaches Fukuoka. Other NOy consists of NO, $NO_2$, $NO_3$, $HNO_2$, $N_2O_5$, and PANs.