# Peer review of "Nitrate transboundary heavy pollution over East Asia in winter"

_Atmospheric Chemistry and Physics, 2016_

## Referee Comment (RC1) · Anonymous Referee #1 · 9 Dec 2016

This study analyzed two episodes (characterized as type N and type S according to the dominant compositions) with the high PM2.5 concentrations reaching around 100 μg/m3 during an intensive observation campaign in January 2015 at Fukuoka in western Japan. Several ground-based measurements and the CMAQ model as well as the path analysis of HYSPLIT model have been utilized to investigate the transboundary air pollution for both types. Authors addressed their results with the comprehensive methods and proved the importance of the transboundary air pollution dominated by NO3-, which will help refine our understanding of the transboundary heavy PM2.5 pollution in winter over East Asia. However, there are several rooms the paper can be much improved scientifically, such as the non-linearity effects of the sensitivity simulation to the secondary pollutions and the explanation of high speed of transboundary air pollution. If we take the 1000 km distance between coastline of China and western Japan, which is assumed by the authors, the transport of air mass speed will be almost 15 m/s while the traveling time is 18h. Is this reasonable for the wind speed reaching so high during the observation period? Overall, this is a nice piece of paper with clear objectives and methods and will provide valuable results. I recommended it for publication in Atmospheric Environment after minor revisions. Some comments and suggestions are listed as follows:

1. On Page 3, Line15. Observation and model simulation section. Authors should introduce their dealing methods for the different data. For example, the chemical compositions of aerosols measured by ACSA-12 and Denuder-filter pack method are 1 hour and 6-8 h, respectively. For CMAQ model, it is the hourly results. So, how could authors get the statistical parameters like R, MFE?

2. On Page 6, Line12. Authors introduced the emission settings in the model simulation. They assumed the emissions in 2008 are similar with that in 2015. Although they issued the NO2 column in China from satellite observation is similar to those for 2009, the SO2 is complicated. How is the picture for SO2 emission? And How about the VOCs? At least, the emission amount for the primary air pollutants between China, Korea and Japan should be listed out.

3. On Page 7, Line3. "Because the amount of emissions from China is larger than that from Japan, to avoid large nonlinearities in the atmospheric concentration response to emissions variation (e.g., Itahashi et al., 2015), the sensitivity simulation was designed to switch off the anthropogenic emissions in Japan." Why the anthropogenic emissions in Japan be switched off could avoid the nonlinearities? What is the amount of the anthropogenic emissions taken up in Japan, and how about the other sources, like biogenic and agriculture? Because, based on the previous study, the emission cut by 20-30% may decrease the nonlinearities in maximum in the sensitivity simulation.

4. On Page 7, Line9. It seems that Fig 2. should be Fig 4. and same as that Fig. 2a to Fig 4a.

5. On Page 7, Line20. Temporal variation of particulate matter. Authors presented very good simulation of particulate matters as well as their compositions in Japan and China during the period. It is curious to me, during the type N and type S episodes, the simulated wind speed is much higher than the observations, how the air pollution simulated well?

6. On Page 8, Line10. "therefore, the transboundary air pollution was dominant during January 2015". First, similar as the above mentioned how the authors delimited the non-linearities just from switch off the anthropogenic emission in Japan? Second, how about the anthropogenic emission take up in the whole emission in Japan, what about the biogenic, such as ocean sources?

7. On Page 10, Line14. "Based on the model results, because the domestic contribution for HNO3 was observed on January 14". It is confused to me that HNO3 was observed since this the model results.

8. On Page 10, Line23. BC section. BC is over estimated during both type N and type S episodes, while SO42- and NH4+ is underestimated. Can the authors explain this? Since in the following sections, "the rates of decrease of total sulfate, total nitrate, and total ammonia were generally consistent with the rates of decrease of normalized BC and CO." (On page 14, Line 26), and "For SO42-, the concentration was higher when the air mass arrived at Fukuoka compared with that in China, suggesting

the fast production of SO42- during the transport process." (On page 12, Line 3), if the BC is over estimated, the SO42- should be more overestimated. One exceptions, the BC or the transboundary has been overestimated in China.

9. On Page 11, Line25. "The traveling time from the coast of China to Fukuoka was about 18 h." As it is mentioned above, the traveling speed will be reached at 15 m/s, which means the wind speed should be 15 m/s. Is this reasonable? From the observations of meteorological conditions in Fukuoka, during the two episodes, the wind speed is 5-8 m/s, which is significant slow than 15 m/s.

---

## Referee Comment (RC2) · Anonymous Referee #2 · 12 Dec 2016

This paper focuses on the nitrate transboundary heavy pollution over East Asia in winter. They developed a technique that could differentiate the nitrogen or sulfur dominated air pollution based on a regional chemical transport model and surface observations. They also highlighted the importance of the transboundary air pollution dominated by nitrate, which may refine our understanding of the transboundary heavy PM2.5 pollution in winter over East Asia. Overall, this represents an important work to document the sources-transport-deposition of air pollutants in a hotspot region. I recommend it to be accepted but with revision, to allow the authors to address my concerns below. Also, they could provide, if possible further analysis from another episode to see how comparable/consistent their results could be, especially for type nitrate episode.

Page 2, Line 18: The conversion of gas to particle is not the only way producing nitrate
in the air. Besides, coarse particles like Gobi desert surface soil also involved in the nitrate production during the transport [Atmos. Chem. Phys., 14, 11571-11585].

Page 3: It is better to introduce "why the observation period from January 7–17, 2015 was selected".

Page 4, Line 12: The citation (Pan et al., 2016) is not identical throughout the text. It is better to use (Pan X. et al., 2016) here, to avoid mixing from another EST paper by Pan Y. et al., 2016.

Page 4, Sect. 2.1.1, 2.1.2 and 2.1.3: All the measurements made have biases and in some cases they are significant. It was reported here that fine-mode aerosols were collected with a PTFE filter. It is well documented that positive artifacts of filter sampling are mainly caused by the adsorption of interference gases, such as acidic HNO3 gases, by the collected particles or the sampling filter [Atmospheric Environment 145, 293–298]. It is likely that nitrates have been overestimated due to the sampling filter itself, in addition to the difference in the cut-off diameter [Page 5, Line 1-4].

Page 5, Line 1-4: Is the difference in the cut-off diameter was the only reason for the systematic differences between ACSA and D-F method? It was documented that ultra-violet spectrophotometric method will overestimate the concentration of nitrate. Please also add some details about the comparison here, as most of the readers cannot follow references in Japanese.

Page 5, Line 13-14: It is easy to understand that the nitrate data from Goto Islands, Tsushima Island, and Tottori was not used due to volatilization. In addition to nitrate, ammonium was also affected by volatilization. So, why the data of ammonium was used?

Page 5, Line 5-22: How did PM2.5 measured by PM-712 compare to the Beta at-tenuation sampler? It is likely that PM2.5 have been overestimated, depending on temperature and relative humidity, and there could have been significant impacts.

Page 5, Line 22: Change the title of 2.1.5 to 2.1.6.

Page 9, Line 10-12: Even that I agree with the results in this study, I am unsure how reproducible those results are on the regular basis, since the authors only study one episode for "type N" and "type S", respectively. Such an issue needs to be critically addressed.

Page 9, Line 25-30: It is hard to believe the concentrations of ammonia are close to zero. As shown in Fig. 6, it seems the temporal variation of the NHx observation was mismatched with the ammonia simulation, especially for Type N. In a recent publication, enhanced values of NH3 were observed within the Asian summer monsoon upper troposphere, where it might contribute to the composition of the Asian tropopause aerosol layer [Atmos. Chem. Phys., 16, 14357-14369].

Page 10, Line 20-21: Why precipitation on Jan 15-16 was not captured in the model [Fig. 4]?

Page 11, Sect. 3.2: Although the long-term range transport of air pollutant was discussed here, the readers are still wondering the contribution of nitrate production during the process/pathway of transport, as well as the contribution from local sources surrounding the investigated sites.

Page 16: The aforementioned contributions are encouraged to be quantified in the conclusion and abstract to highlight the importance of nitrate long-range transport.
* * *

---

## Referee Comment (RC3) · Anonymous Referee #3 · 12 Dec 2016

This manuscript deals with an episodic model simulation and evaluation with PM2.5 and its composition measurements. the relationship among sulfate, nitrate and ammonium and the neutralization degree are well explained and discussed with the observations and simulation results. The manuscript provides interesting and scientific information on PM2.5 formation and transport in Northeast asia. However it is not clear what the authors want to tell in the manuscript. The contents are good enough, but it would be better reconsider the discussion points. Specific comments are listed below.

Abstract, Lines 29∼35, "Analyzing the gas ratio, which is an indicator of the sensitivity of NO3- to changes in SO42- and NH4+, showed that the air mass over China was super NH3-rich for type N, but was almost NH3-neutral for type S. ∼"

1) Is it because of low SO2 to Sulfate conversion rate in Type N? It should be also

noted that the observed levels of total NH3 in Figure 6 are almost identical for Type N and Type S.

2) Why are NH3 conditions for Type S and Type N different when we consider the target modeling period is quite short?

3) What determines Type N and Type S? Is it due to meteorology, different origins of air plumes, or pathways of back-trajectories? It seems that outflow from Shanghai area affects the monitoring sites for Type N, and air plume is transported from Hebei and Beijing for the Type S.

Figure 5, Why domestic influence does not appear on January 13 and 14 when dominant wind direction shown on Figure 4(c) is easterly?

Page 9, Lines 10: "the main component of SNA was NO3- during the first episode and SO42- during the second episode." –> It would be helpful to indicate quantitative portions of nitrate and sulfate for the episodes.

Page 9, Line 20: " However, 20 the sensitivity simulation confirmed that the transboundary NO3- air pollution was dominant for types N and S." –> The sentence is not clear. Does it mean that Type N and Type S are determined by sulfate concentration or Fs rather than nitrate concentrations?

Trajectory analyses: It seems trajectory results should be checked. For the Tn case in Figure 9, the trajectory and wind vectors are relatively well matched. However, in Figure 10, more northerly wind is dominant, and the trajectory origin should move more northward. But, tracjectoies in Figures 9 and 10 are almost the same.

---

## Author Comment (AC1) · 27 Jan 2017

Response to Reviewer #1

This study analyzed two episodes (characterized as type N and type S according to the dominant compositions) with the high PM2.5 concentrations reaching around 100 μg/m3 during an intensive observation campaign in January 2015 at Fukuoka in western Japan. Several ground-based measurements and the CMAQ model as well as the path analysis of HYSPLIT model have been utilized to investigate the transboundary air pollution for both types. Authors addressed their results with the comprehensive methods and proved the importance of the transboundary air pollution dominated by NO3-, which will help refine our understanding of the transboundary heavy PM2.5 pollution in winter over East Asia. However, there are several rooms the paper can be much improved scientifically, such as the non-linearity effects of the sensitivity simulation to the secondary pollutions and the explanation of high speed of transboundary air pollution. If we take the 1000 km distance between coastline of China and western Japan, which is assumed by the authors, the transport of air mass speed will be almost 15 m/s while the traveling time is 18h. Is this reasonable for the wind speed reaching so high during the observation period? Overall, this is a nice piece of paper with clear objectives and methods and will provide valuable results. I recommended it for publication in Atmospheric Environment after minor revisions. Some comments and suggestions are listed as follows:

Dear Reviewer #1

Thank you for taking the time to review our manuscript for *Atmospheric Chemistry and Physics*, and providing helpful comments. To address your comments on the non-linearity of effects involved in this study, we have revised and added explanations. For the wind speed, we think we have fully addressed your concerns.
We have revised our manuscript according to the reviewers' comments and suggestions. We believe that these revisions address all points raised by the reviewers. We have also provided a point-by-point response below. The revisions are indicated in blue in the revised manuscript.

Sincerely,

Syuichi Itahashi

1) On Page 3, Line15. Observation and model simulation section. Authors should introduce their dealing methods for the different data. For example, the chemical compositions of aerosols measured by ACSA-12 and Denuder-filter pack method are 1 hour and 6-8 h, respectively. For CMAQ model, it is the hourly results. So, how could authors get the statistical parameters like R, MFE?

> The statistical analysis for $PM_{2.5}$ discussed in Page 8, Lines 7-9 and Lines 25-26 were, respectively, based on ACSA-12 and BAMs. The measurement interval is 1 h. We have compared these observational data directly to CMAQ model output data.

2) On Page 6, Line12. Authors introduced the emission settings in the model simulation. They assumed the emissions in 2008 are similar with that in 2015. Although they issued the NO2 column in China from satellite observation is similar to those for 2009, the SO2 is complicated. How is the picture for SO2 emission? And How about the VOCs? At least, the emission amount for the primary air pollutants between China, Korea and Japan should be listed out.

> $SO_2$ emission might be overestimated because of the assumption about the emission level in the year 2008. Because the appropriate reference for VOC is not available, the emissions amount of VOC is assumed to be at the 2008 level. In this study, our focus was on the behavior of sulfate-nitrate-ammonium (SNA). We have listed the emissions amounts for $SO_2$, NOx, and $NH_3$ in Table 1, which is newly included. We have added a brief comment about this issue in Section 2.2.

3) On Page 7, Line3. "Because the amount of emissions from China is larger than that from Japan, to avoid large nonlinearities in the atmospheric concentration response to emissions variation (e.g., Itahashi et al., 2015), the sensitivity simulation was designed to switch off the anthropogenic emissions in Japan." Why the anthropogenic emissions in Japan be switched off could avoid the nonlinearities? What is the amount of the anthropogenic emissions taken up in Japan, and how about the other sources, like biogenic and agriculture? Because, based on the previous study, the emission cut by 20-30% may decrease the nonlinearities in maximum in the sensitivity simulation.

> We agree that the method with an emissions cut by 20-30% (e.g., Fiore et al., 2009, J. of Geophys. Res. 114: D04301) is a suitable approach, but for simplicity, we have applied the zero-out method in this study. We have fully revised the relevant sentence as follows (P7, L21-25).
>
> "In terms of $O_3$, which is involved in complex nonlinear chemistry, larger nonlinearities in the atmospheric concentration response to emissions variation for China but not Japan were clarified due to the higher amount of emissions from China than from Japan (Itahashi et al.,

2015). Therefore, the sensitivity simulation was designed to remove anthropogenic emissions in Japan instead of those in China."

The term 'anthropogenic' indicates that the emissions were taken from the REAS inventory; because of this, the agriculture category was included but the biogenic category was not. We have also added an explanation of the treatment of anthropogenic emissions as follows (P7, L19-20).

"Here, anthropogenic emissions were taken from the REAS inventory; because of this, emissions from agriculture were included."

4) On Page 7, Line9. It seems that Fig 2. should be Fig 4. and same as that Fig. 2a to Fig 4a.

This mislabeling has been corrected. We have also inserted the appropriate figure number for the other meteorological parameters.

5) On Page 7, Line20. Temporal variation of particulate matter. Authors presented very good simulation of particulate matters as well as their compositions in Japan and China during the period. It is curious to me, during the type N and type S episodes, the simulated wind speed is much higher than the observations, how the air pollution simulated well?

The wind speed at Fukuoka, Japan, is overestimated compared with observations, as we have shown in Fig. 4b. This is partly related to the land use mapping at 27-km resolution. The grid corresponded to Fukuoka is assigned to the land category; however, the surrounding north-west grid is assigned to the ocean category (Supplemental Fig. 1-1). Therefore, the observed slowed wind speed, which arises from the large effect of friction over land, might not be simulated well. Please also see reply 9).

[Figure]

Supplemental Figure 1-1. Mapping of land (brown) and ocean (light blue) categories around Kyushu island. The red square indicates the grid square of Fukuoka.

6) On Page 8, Line10. "therefore, the transboundary air pollution was dominant during January 2015". First, similar as the above mentioned how the authors delimited the non-linearities just from switch off the anthropogenic emission in Japan? Second, how about the anthropogenic emission take up in the whole emission in Japan, what about the biogenic, such as ocean sources?

First, this sentence mentioned the transboundary air pollution status at Goto, Tsushima, and Tottori. We have added this point (P8, L30-31) as "at remote sites in western Japan". At Fukuoka, we found domestic contribution in some cases. Because the zero-out method was applied to the emissions of Japan instead of those of China, nonlinearity will be smaller in the case of the zero-out method for Chinese emissions. To support a discussion about nonlinearity of the relevant chemistry, we further used a BC variation to investigate the local and transboundary contributions. Considering that BC variation, the dominance of the transboundary air pollution on both types can be assumed.

Second, only anthropogenic emissions were switched off in this case; the biogenic emissions were not switched off. As a biogenic source, dimethylsulfide (DMS) emissions from oceans were not included in the modeling system. We have explicitly mentioned the treatment of DMS (P7, L10-11).

7) On Page 10, Line14. "Based on the model results, because the domestic contribution for HNO3 was observed on January 14". It is confused to me that HNO3 was observed since this the model results.

This discussion was based on model results. To avoid a misreading, we have changed the wording from "observed" to "found."

8) On Page 10, Line23. BC section. BC is over estimated during both type N and type S episodes, while SO42- and NH4+ is underestimated. Can the authors explain this? Since in the following sections, "the rates of decrease of total sulfate, total nitrate, and total ammonia were generally consistent with the rates of decrease of normalized BC and CO." (On page 14, Line 26), and "For SO42-, the concentration was higher when the air mass arrived at Fukuoka compared with that in China, suggesting the fast production of SO42- during the transport process." (On page 12, Line 3), if the BC is over estimated, the SO42- should be more overestimated. One exceptions, the BC or the transboundary has been overestimated in China.

Because BC concentrations are changed via emission/deposition/transport processes, but SNA concentration are also involved in the chemistry, the model tendency to overestimate concentration is not necessarily related. The reason for model overestimation of BC at Goto might be related to the assumption about BC emissions in China.

9) On Page 11, Line25. "The traveling time from the coast of China to Fukuoka was about 18 h."
As it is mentioned above, the traveling speed will be reached at 15 m/s, which means the wind
speed should be 15 m/s. Is this reasonable? From the observations of meteorological conditions in
Fukuoka, during the two episodes, the wind speed is 5-8 m/s, which is significant slow than 15 m/s.

As we mention in reply 5), the wind speed at Fukuoka is slowed by friction over land. In
Supplemental Figures 1-2 and 1-3, wind speeds are shown for types N and S, respectively. In
both types, episode-averaged wind speed over the Yellow Sea ranged from 8 to 12 m/s, which
is greater than the observed wind speed at Fukuoka. Before the air mass arrived at Fukuoka,
wind speed was further increased, beyond 12–16 m/s, over the eastern part of the East China
Sea.

Type N

[Figure]

Supplemental Figure 1-2. Wind speed during type N episode (a) averaged over whole episode,
and (b) averaged 3 h before the air mass reached Fukuoka.

Type S
(a) averaged over whole episode          (b) before the air mass reached Fukuoka

[Figure]

wind speed [m/s]

Supplemental Figure 1-3. Wind speed during type S episode (a) averaged over whole episode, and (b) averaged 3 h before the air mass reached Fukuoka.

---

## Author Comment (AC2) · 27 Jan 2017

Response to Reviewer #2

This paper focuses on the nitrate transboundary heavy pollution over East Asia in winter. They developed a technique that could differentiate the nitrogen or sulfur dominated air pollution based on a regional chemical transport model and surface observations. They also highlighted the importance of the transboundary air pollution dominated by nitrate, which may refine our understanding of the transboundary heavy PM2.5 pollution in winter over East Asia. Overall, this represents an important work to document the sources-transport-deposition of air pollutants in a hotspot region. I recommend it to be accepted but with revision, to allow the authors to address my concerns below. Also, they could provide, if possible further analysis from another episode to see how comparable/consistent their results could be, especially for type nitrate episode.

Dear Reviewer #2,

Thank you for taking the time to review our manuscript for *Atmospheric Chemistry and Physics*, and providing helpful comments. We agreed that more case studies are necessary to figure out the transboundary air pollution of nitrate. In this study, we have focused on an intensive observation period in January, and comprehensively analyzed two episodes. We would like to continue to apply model analysis to other episodes and report those results on another occasion.

We have revised our manuscript according to the reviewers' comments and suggestions. We believe that these revisions address all points raised by the reviewers. We also provide a point-by-point response below. The revisions are indicated in blue in the revised manuscript.

Sincerely,

Syuichi Itahashi

1) Page 2, Line 18: The conversion of gas to particle is not the only way producing nitrate in the air. Besides, coarse particles like Gobi desert surface soil also involved in the nitrate production during the transport [Atmos. Chem. Phys., 14, 11571-11585].

In this sentence, we have mentioned $NO_3^-$ as fine mode. To clarify, we have revised the wording of '$NO_3^-$' to '$NO_3^-$ in $PM_{2.5}$'.

The coarse particles of $NO_3^-$ produced via the reaction with mineral dust are considered to be smaller in the winter case.

2) Page 3: It is better to introduce "why the observation period from January 7–17, 2015 was selected".

To capture the heavy transboundary air pollution in winter time, we have set an intensive observation for this period. This was introduced in the text (P3, L10-11).

3) Page 4, Line 12: The citation (Pan et al., 2016) is not identical throughout the text. It is better to use (Pan X. et al., 2016) here, to avoid mixing from another EST paper by Pan Y. et al., 2016.

Thank you for careful reading. We have modified these citations to indicate the appropriate references.

4) Page 4, Sect. 2.1.1, 2.1.2 and 2.1.3: All the measurements made have biases and in some cases they are significant. It was reported here that fine-mode aerosols were collected with a PTFE filter. It is well documented that positive artifacts of filter sampling are mainly caused by the adsorption of interference gases, such as acidic HNO3 gases, by the collected particles or the sampling filter [Atmospheric Environment 145, 293–298]. It is likely that nitrates have been overestimated due to the sampling filter itself, in addition to the difference in the cut-off diameter [Page 5, Line 1-4].

Because the gas-phase $HNO_3$ was collected with an annular denuder coated with NaCl before collection in a PTFE filter, the possibility of the adsorption of $HNO_3$, which would lead to overestimation of $NO_3^-$, is considered negligible. We have added an explanation of this (P5, L4-5).

5) Page 5, Line 1-4: Is the difference in the cut-off diameter was the only reason for the systematic differences between ACSA and D-F method? It was documented that ultraviolet spectrophotometric method will overestimate the concentration of nitrate. Please also add some details about the comparison here, as most of the readers cannot follow references in Japanese.

The difference in the cut-off diameter was considered to be the most important factor in differences between results from the ACSA and D-F methods. To support this assertion, we have revised our explanation (P5, L7-14), which is reproduced here.

"Comparing size-segregated $SO_4^{2-}$ and $NO_3^-$ data based on the D-F pack method with that from ACSA showed systematic differences. Linear regression analysis of fine-mode and coarse-mode $SO_4^{2-}$ and $NO_3^-$ showed good correlation, but the slope was not unity. Fine-mode aerosols were underestimated by the D-F pack relative to ACSA measurements, and coarse-mode aerosols were overestimated by the D-F pack relative to ACSA measurements. However, by summing the fine- and coarse-mode data, the slope between D-F pack and ACSA results was close to unity. The difference in the cut-off diameter of the D-F pack method was considered to be the most important factor in explaining differences between results from D-F pack and ACSA. More details of the comparison and validation of ACSA data are reported in Osada et al. (2016)."

Page 5, Line 13-14: It is easy to understand that the nitrate data from Goto Islands, Tsushima Island, and Tottori was not used due to volatilization. In addition to nitrate, ammonium was also affected by volatilization. So, why the data of ammonium was used?

We agree that ammonium particles can be somewhat affected by volatilization in the case of ammonium-nitrate formation, but ammonium can form ammonium-sulfate in preference, and the volatilization of ammonium-sulfate will not be considered. Therefore, we used ammonium concentration, which captures ammonium-sulfate. To note this point to the readers, we have mentioned the possibility of the volatilization of ammonium-nitrate (P5, L24-26).

Page 5, Line 5-22: How did PM2.5 measured by PM-712 compare to the Beta attenuation sampler? It is likely that PM2.5 have been overestimated, depending on temperature and relative humidity, and there could have been significant impacts.

PM-712 data were acquired at remote sites in Japan (Section 2.1.4), and beta attenuation sampler data were used at the observation sites in China (Section 2.1.5). We did not directly compare these results.

Page 5, Line 22: Change the title of 2.1.5 to 2.1.6.

We have revised the numbering.

Page 9, Line 10-12: Even that I agree with the results in this study, I am unsure how reproducible those results are on the regular basis, since the authors only study one episode for "type N" and "type S", respectively. Such an issue needs to be critically addressed.

Because of the limitation inherent in a short-term intensive observation campaign, we have discussed one episode of 'Type N' and one episode of 'Type S' in this study. We would like to

further analyze and demonstrate transboundary air pollution cases for both types; however, such analyses are beyond the scope of this study.

Page 9, Line 25-30: It is hard to believe the concentrations of ammonia are close to zero. As shown in Fig. 6, it seems the temporal variation of the NHx observation was mismatched with the ammonia simulation, especially for Type N. In a recent publication, enhanced values of NH3 were observed within the Asian summer monsoon upper troposphere, where it might contribute to the composition of the Asian tropopause aerosol layer [Atmos. Chem. Phys., 16, 14357-14369].

As an observation result, $NH_3$ showed nearly zero concentration for both types and in some cases (Fig. 6). As we have discussed in the text (P10, L15-19), $NH_3$ was fully converted to produce $NH_4^+$ in both types.

The reference mentioned here discussed the tropopause aerosol layer during the Asian summer monsoon, and this would not be closely related to our results, which concern the air quality in winter.

Page 10, Line 20-21: Why precipitation on Jan 15-16 was not captured in the model [Fig. 4]?

At 1500 LT, 15 January, precipitation at 2.5 mm/h was observed. For this precipitation event, the model underestimated the precipitation amount as 0.22 mm/h. Over north Kyushu island, the model simulated less than 1 mm/h of precipitation for this event. In general, the model simulation underestimated precipitation amounts, as we have reported in a previous study (Itahashi et al., 2014, Atmospheric Environment, 92: 171-177).

Page 11, Sect. 3.2: Although the long-term range transport of air pollutant was discussed here, the readers are still wondering the contribution of nitrate production during the process/pathway of transport, as well as the contribution from local sources surrounding the investigated sites.

First, nitrate production during transport from China to Fukuoka did not occurred in the examined winter cases, as we show in Figs. 11 and 12 for the trajectories and summarize in Fig. 13. In both types, high concentration of nitrate was directly transported from China, and the concentration level gradually decreased during transport (Fig. 13). We demonstrated that nitrate can be directly transported if sulfate production is not activated (Type N, Fig. 11), and nitrate concentration gradually decreased under the production of sulfate-ammonium (Type S, Fig. 12).

Second, as we discuss in Section 3.2, both types N and type S were dominated by the transboundary air pollution, and local contribution was not considered. Nitrate production involves nonlinearity with complex chemical mechanisms, but observation and model of black carbon also suggested little local contribution for both types.

Page 16: The aforementioned contributions are encouraged to be quantified in the conclusion and abstract to highlight the importance of nitrate long-range transport.

To address the comment, we have added a discussion of the direct transport of nitrate from China to Fukuoka (P15, L22-23).

Additionally, we have added a comment to the abstract about the high concentration of $NO_3^-$ over China to emphasize the direct transport of $NO_3^-$ from China to Japan.

---

## Author Comment (AC3) · 27 Jan 2017

Response to Reviewer #3

This manuscript deals with an episodic model simulation and evaluation with PM2.5 and its composition measurements. the relationship among sulfate, nitrate and ammonium and the neutralization degree are well explained and discussed with the observations and simulation results. The manuscript provides interesting and scientific information on PM2.5 formation and transport in Northeast asia. However it is not clear what the authors want to tell in the manuscript. The contents are good enough, but it would be better reconsider the discussion points. Specific comments are listed below.

Dear Reviewer #3,

Thank you for taking the time to review our manuscript for *Atmospheric Chemistry and Physics*, and providing helpful comments.
We have revised our manuscript according to the reviewers' comments and suggestions. We believe that these revisions address all points raised by the reviewers. We also provide a point-by-point response below. The revisions are indicated in blue in the revised manuscript.

Sincerely,

Syuichi Itahashi

Abstract, Lines 29-35, "Analyzing the gas ratio, which is an indicator of the sensitivity of NO3- to changes in SO42- and NH4+, showed that the air mass over China was super NH3-rich for type N, but was almost NH3-neutral for type S."

1) Is it because of low SO2 to Sulfate conversion rate in Type N? It should be also noted that the observed levels of total NH3 in Figure 6 are almost identical for Type N and Type S.

In the two cases analyzed in this study, the decomposition of $NO_3^-$ was determined by the production of $SO_4^{2-}$. The decomposition rate of $NO_3^-$ and the production rate of $SO_4^{2-}$ during transport from China to Fukuoka are marked in Fig. 13.

2) Why are NH3 conditions for Type S and Type N different when we consider the target modeling period is quite short?

Because the backward trajectories were similar in both types, $NH_3$ emission intensity is not a factor in governing the $NH_3$ conditions. Due to the short lifetime of $NH_3$ in the atmosphere, $NH_3$ condition can change within the target period.

3) What determines Type N and Type S? Is it due to meteorology, different origins of air plumes, or pathways of back-trajectories? It seems that outflow from Shanghai area affects the monitoring sites for Type N, and air plume is transported from Hebei and Beijing for the Type S.

The meteorological conditions that affect $SO_4^{2-}$ production are one factor in determining type N and S in our case. Because the backward trajectories were similar in both types, the origins of the air mass and the related emission intensity would not be important factors.

Although we can find a high concentration around Shanghai in Fig. 9, the air masses from Hebei and Shandong provinces were mainly transported along the backward trajectory of $T_N$ from China to Fukuoka, as shown by the analysis of temporal variation of spatial distribution.

Figure 5, Why domestic influence does not appear on January 13 and 14 when dominant wind direction shown on Figure 4(c) is easterly?

On 13 and 14 January, domestic contributions for nitrate were found at Fukuoka (Fig. 6). With regard to $SO_4^{2-}$, the local contribution at Fukuoka was small through the year, as we showed in a previous study (Itahashi et al., 2017) and as was also suggested from observation (Kaneyasu et al., 2014). Over the remote islands of Goto and Tsushima, and the remote site of Tottori, no local contribution was found, which we attribute to the low amount of $SO_2$ emissions.

Page 9, Lines 10: "the main component of SNA was NO3- during the first episode and SO42- during the second episode." –> It would be helpful to indicate quantitative portions of nitrate and sulfate for the episodes.

Quantitative portions have been added in the revised manuscript. We have added the following sentences (P9, L33-P10, L1).

"At Fukuoka, the relative portions of $SO_4^{2-}$, $NO_3^-$, and $NH_4^+$ within $PM_{2.5}$ were respectively 18%, 20%, and 14% during the first episode, and 27%, 6%, and 14% during the second episode."

Page 9, Line 20: " However, 20 the sensitivity simulation confirmed that the transboundary NO3- air pollution was dominant for types N and S." –> The sentence is not clear. Does it mean that Type N and Type S are determined by sulfate concentration or Fs rather than nitrate concentrations?

For the two cases analyzed in this study, nitrate was directly transported from China and decomposed into gas-phase during transport from China to Fukuoka, which was clarified in Fig. 13. The decomposition rate can be determined from the sulfate concentration.

Trajectory analyses: It seems trajectory results should be checked. For the Tn case in Figure 9, the trajectory and wind vectors are relatively well matched. However, in Figure 10, more northerly wind is dominant, and the trajectory origin should move more northward. But, tracjectoies in Figures 9 and 10 are almost the same.

Because the wind fields on Figs. 9 and 10 were thinned for better visualization, some differences between Type N and S are displayed. We have checked the backward trajectories again and confirmed that the transport pattern is similar between types. The original figures for trajectory analysis are shown as Supplemental Figure 3-1.

[Figure]

Supplemental Figure 3-1. Backward trajectory for (left) Type N, and (right) Type S.

---

## Editor Decision (ED1)

Acrobat X または Adobe Reader X 以降でこの PDF ポートフォリオを開いてこれまでにない便利さを体験してください。

Adobe Reader を今すぐダウンロード!

---

## Author Response (AR2)

Acrobat X または Adobe Reader X 以降でこの PDF ポートフォリオを開いてこれまでにない便利さを体験してください。

Adobe Reader を今すぐダウンロード!

---

## Author Response (AR3)

Dear Editor

Thank you for taking the time to deal with our manuscript for *Atmospheric Chemistry and Physics*, and providing helpful comments.

According to your comment, we have revised the caption of Fig. 2, and related text. The revisions are indicated in blue in the revised manuscript.

Regarding your concerns for English, our manuscript has been checked by native English speaker. The certification of English editing service was attached on the end of manuscript.

Sincerely,

Syuichi Itahashi

[revised manuscript text omitted]

February 8, 2017

**Certificate of Editing**

To whom it may concern:

This letter certifies that Dr. Syuichi Itahashi's manuscript entitled "Nitrate transboundary heavy pollution over East Asia in winter" has been professionally edited by a native speaker familiar with the author's field. Assuming that the author has made no additional changes to the text, the manuscript should meet the grammatical standards required by an English-language publication.

ThinkSCIENCE K.K. is a Japan-based company specializing in providing high-quality editing, proofreading, and translation services to Japanese authors in all fields of science, medicine and technology. We also support the work of academic journals and societies, and corporations. If you would like more information about our services or policies, please feel free to contact us.

Sincerely,

John Zepernick
Managing Editor
ThinkSCIENCE, Inc.